# Evaluation of cutaneous immune response in a controlled human in vivo model of mosquito bites

Mosquito-borne viruses are a growing global threat. Initial viral inoculation occurs in the skin via the mosquito 'bite', eliciting immune responses that shape the establishment of infection and pathogenesis. Here we assess the cutaneous innate and adaptive immune responses to controlled *Aedes aegypti* feedings in humans living in *Aedes*-endemic areas. In this single-arm, cross-sectional interventional study (trial registration #NCT04350905), we enroll 30 healthy adult participants aged 18 to 45 years of age from Cambodia between October 2020 and January 2021. We perform 3-mm skin biopsies at baseline as well as 30 min, 4 h, and 48 h after a controlled feeding by uninfected *Aedes aegypti* mosquitos. The primary endpoints are measurement of changes in early and late innate responses in bitten vs unbitten skin by gene expression profiling, immunophenotyping, and cytokine profiling. The results reveal induction of neutrophil degranulation and recruitment of skin-resident dendritic cells and M2 macrophages. As the immune reaction progresses T cell priming and regulatory pathways are upregulated along with a shift to $T_h2$-driven responses and CD8[+] T cell activation. Stimulation of participants' bitten skin cells with *Aedes aegypti* salivary gland extract results in reduced pro-inflammatory cytokine production. These results identify key immune genes, cell types, and pathways in the human response to mosquito bites and can be leveraged to inform and develop novel therapeutics and vector-targeted vaccine candidates to interfere with vector-mediated disease.

With approximately one million deaths every year, vector-borne diseases remain a major health problem worldwide[1]. Arboviruses are of particular concern given a global distribution and a burden of nearly half a billion infections[2]. Several clinically significant arboviruses such as yellow fever (YFV), dengue (DENV), chikungunya (CHIKV), and Zika (ZIKV) viruses are all transmitted by *Aedes* spp. mosquitoes. These vectors are abundant in tropical and subtropical areas, but current climate change and rapid urbanization are expanding their transmission potential and increasing the at-risk population[3].

Mosquitoes first acquire an arbovirus via insertion of their proboscis into the dermis of an infected host during a blood meal. The virus then replicates in midgut cells to later disseminate and reach the salivary glands (SG)[4]. During a subsequent blood meal, the infectious mosquito inoculates virus and insect-derived molecules, including salivary components, into the dermis of a new human host. The saliva from hematophagous mosquitoes is a cocktail of pharmacologically active molecules, some of which are anti-hemostatic, angiogenic, and immunomodulatory[5–7]. Over the last couple of decades, several animal and in vitro studies demonstrated that inoculation of vector saliva, and/or concomitant blood-feeding by an arthropod, can immunomodulate the host antiviral response in the skin and periphery[7–10]. These perturbations result in altered cytokine production profiles[11–13], promotion of recruitment of infection-susceptible cells to the bite site[14,15], stimulation of autophagy activation[5], and an induction of neutrophil

✉ e-mail: jessica.manning@nih.gov

**Fig. 1 | Graphical representation of study design.** Thirty Cambodian participants were enrolled in the study. Each patient underwent a mosquito feeding (five female *Aedes aegypti*) for 10 min. Each volunteer underwent a 3-mm biopsy from a mosquito bite site at 30 min, 4 h, and 48 h. Distinct bite sites were biopsied at different time points. Additionally, two 3-mm biopsies of normal skin (NSK) were taken from the opposite forearm at baseline. Blood was collected to obtain serum. Designed with Biorender.

infiltration to the local inflammation site[15] amongst other consequences. Mosquito-inoculated DENV or ZIKV infection in macaques led to higher or delayed viremia compared to non-vector or needle inoculation routes[16,17]. With regard to clinical pathogenesis, mice inoculated with both Semliki Forest virus and mosquito saliva had higher loads of virus RNA in the skin, earlier brain dissemination, and more frequent lethal outcomes than animals inoculated with virus alone[12]. Taken together, arthropod-mediated viral infection leads to increased disease progression, viremia, and mortality[8,9]. Despite this robust evidence from animal and in vitro studies, there are no translational studies on the human skin response to mosquito bites, much less those in tropical areas. Individuals are continuously exposed to mosquito bites, possibly shaping their immune responses to mosquito salivary proteins in contrast to those intermittently or rarely exposed[18,19]. In order to examine the dermal and epidermal innate and adaptive immune responses to *Ae. aegypti* saliva, we compared the immune responses before and after *Ae. aegypti* bites in skin biopsies from 30 healthy Cambodian individuals living in an area where *Ae. aegypti* mosquitoes are prevalent. Our study provides information on the key genes and cell populations involved in the human immune response at different time points after the 'mosquito bite.'

## Results

### Controlled *Aedes aegypti* bite challenge in healthy volunteers in Cambodia
From October 2020 to January 2021, we enrolled 30 healthy Cambodian adults at Kampong Speu District Referral Hospital in Chbar Mon, Cambodia. The study population comprised 13 females and 17 males, all of whom were of Khmer (Cambodian) ethnicity (Supplementary Table 1). The population was randomly divided in three groups to assess immune responses by different technologies (Fig. 1). The median age was 32.5 years (IQR 22–35) and the primary occupations were variable, although nearly half (14/30) were unemployed. In terms of

mosquito exposure risk, most participants reported living in a house (20/30), being middle income (25/30), and having one to two domestic water containers (22/30). Notably, most participants did not use larvicide (28/30) or insecticide (19/30) in their homes.

Only 10% (3/30) reported prior dengue infection, but all 30 participants were positive for dengue anti-NS1 antibodies. All participants had detectable IgG to *Ae. aegypti* salivary gland extract (SGE) prior to mosquito feeding, and these levels were not significantly different by age, gender, or 14 days after the mosquito feeding (Supplementary Fig. 1). One participant did not complete Day 14 follow-up and was considered lost to follow-up. No unanticipated adverse events occurred with the biopsies or mosquito feedings.

### Marked clinical changes occurred in the skin post-mosquito feeding
Exposure to five *Ae. aegypti* mosquitos resulted in 3 to 10 visible bite sites per volunteer. Upon clinical observation, participants experienced erythema and swelling with a 'bite size' reaction that peaked at 15 min after feeding (Fig. 2a). The mean bite size of 4.9 mm at 15 min after feeding decreased to 4.7 mm at 30 min, to 2.7 mm at 4 h and gradually decreased to 2.1 mm by 48 h (Supplementary Table 1). Observation of representative histological sections shows increased recruitment of cells to the dermis and epidermis at 48 h post-bite (Fig. 2b, c). There was no difference in bite size reaction by gender or age.

### Changes in gene expression in *Aedes aegypti* bitten versus normal skin (NSK) evolved over 48 h
We extracted RNA and performed next generation sequencing on 40 skin biopsies from 10 participants (baseline normal skin (NSK) biopsy on the forearm, plus 3 follow-up biopsies each at 30 min, 4 h, and 48 h post *Ae. aegypti* bites on the contralateral arm). Principal Component (PC) analysis of *Ae. aegypti* bite site skin biopsies resulted

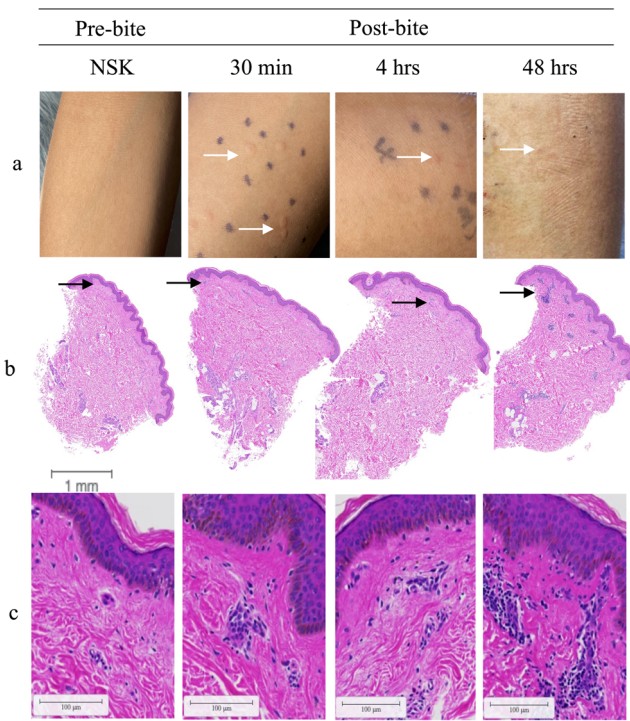

**Fig. 2 | Clinical observations of skin reaction pre- and post-bite on gross examination and H&E staining. a** Skin immune responses to mosquito bites (left to right) normal skin (NSK), 30 min, 4 h, and 48 h after exposure to five *Ae. aegypti* mosquitoes. **b** Representative cross-sectional of 3-mm punch biopsies H&E stained at (left to right) 0 h, 30 min, 4 h, and 48 h after exposure to *Ae. aegypti* bites. Distinct bite sites were biopsied at different time points. Scale bar is 1 mm. Black arrows indicate area of infiltrate. **c** Magnification of infiltrate indicated by black areas in B, scale bar 100 μM.

in segregation of sample clusters based on the time the biopsy was taken (Fig. 3a), with PC1 and PC2 variances defining differences between NSK and 30 min versus 4 h and 48 h. While PC3 variance associated to the separation between biopsies taken at 4 h and 48 h, samples taken from NSK (pink circles) and 30 min after *Ae. aegypti* bites clustered in the same spatial area. At 4 h post-bite, 6 of 10 samples occupied a distinct region (Fig. 3a, blue ellipse) from NSK, 30 min or 48 h after *Ae. aegypti* bites. Similarly at 48 h post-bite, 7 of 10 samples segregated into a discrete space (Fig. 3a, orange ellipse) compared to NSK, 30 min or 4 h after *Ae. aegypti* bites. These data indicate that time post-bite rather than inter-individual differences drive unique patterns in gene expression. Differential analysis of RNA expression with a threshold of two-fold ($-1 <$ log2 fold change (FC) $> 1$; False discovery rate (FDR) $< 0.05$) over NSK biopsies resulted in 709 upregulated genes post-bite compared to 31 downregulated genes (Fig. 3b, red circles). At 30 min, 4 h, and 48 h after a mosquito bite, 26, 210 and 473 genes were upregulated over NSK, respectively (Fig. 3c, Supplementary Data 1). Conversely, 4 genes at 4 h and 27 genes at 48 h were downregulated after *Ae. aegypti* bites (Supplementary Data 1).

Upregulated genes at each time point were subjected to over-representation analysis (ORA) for enriched pathways using the Reactome database with an FDR cut off $< 0.05$. Four pathways related to cell proliferation, cell growth and nuclear transcription were identified (FDR $< 0.05$) at 30 min after *Ae. aegypti* bites (Supplementary Data 2). At both 4 and 48 h post-bite, the immune nature of the host response to *Ae. aegypti* bites was evident via the revealed pathways (Fig. 4A, Supplementary Data 2). At 4 h post-bite, pathways such as "neutrophil degranulation", "Interleukin-4 and Interleukin-13 signaling", "Interleukin-10 signaling", "Interferon-γ signaling", "Chemokine receptors bind chemokines", and "Immunoregulatory interactions between a

Lymphoid and a non-Lymphoid cell" (Fig. 4A, upper chord) were significantly overrepresented and maintained until 48 h post-bite (Fig. 4A, lower chord). Unique pathways discovered at 4 h post-bite were related to "degradation of extracellular matrix" (Fig. 4A, upper chord) while "antimicrobial peptides" pathway and "adaptive immune system processes" were present at 48 h post-bite including "TCR signaling", "Co-stimulation by the CD28 family", "PD-1 signaling", "MHC class II antigen presentation" (Fig. 4A, lower chord). A heatmap with differentially expressed genes (DEG) under an FDR cutoff $< 0.05$ and log2FC $> 1$ in at least one time point was assembled (Supplementary Fig. 2, Supplementary Data 3), showing the dynamics of DE expression throughout the three time points. Figure 4B shows DEG driving the common pathways found on Fig. 4A.

Overall, few DEG genes were detected at 30 min post-bite compared to NSK biopsies. The main upregulated gene *FOSB* (up 18.8x) is one of the transcription factors of the FOS family proteins that regulate cell proliferation and differentiation (Supplementary Data 1). As the reaction progresses over time, gene expression increased towards neutrophil recruitment followed by signaling of both $T_h1$, $T_h2$, and regulatory pathways. The five most regulated genes at 4 h compared to NSK are *KRT6C* (related to keratinization; up 80.8x), *CXCL8* (neutrophil-attracting chemokine *IL-8* up 75.0x), *TNIP3* (inhibitor of NF-κB activation; up 59.3x), *IL-20* (stimulates keratinocyte proliferation and differentiation; up 59.1x) and *IL-1B* (pro-inflammatory cytokine; up 45.7x) (Supplementary Data 1). The immune response evolved at 48 h post-bite to include adaptive immune response pathways related to T-cell priming. The five top genes at 48 h are *KRT6C* (related to keratinization; up 59.6x), *DEFB4A* (an antimicrobial peptide; up 32.7x), *GZMB* (Granzyme B, up 21.1x), *TCL1A* (proliferation of T and B cells; up 20.3x) and *CCL18* (chemokine associated with $T_h2$ responses; up 18.5x).

### Early innate immune response in the skin is characterized by increased proportions of DCs and M2 macrophages and a decreased proportion of NK cells

To confirm and extend the gene expression analysis, dissociated cells from NSK and skin biopsies taken at 30 min and 4 h after exposure to mosquitos were characterized for innate and adaptive immune subsets by flow cytometry (Supplementary Fig. 3, Supplementary Table 2). No significant changes in cell populations were detected between NSK and 30 min post-bite, mirroring the transcriptomic analysis of DEG. At 4 h post-bite, the proportion of dendritic cells (DC) and M2 macrophages increased, while the proportion of mast cells, natural killer (NK), and T cells decreased, when compared to NSK and 30 min post-bite (Fig. 5a). The frequencies of DC subsets increased at 4 h post-bite compared to NSK and 30 min (Fig. 5b–e). Within the DC subsets, Langerhans cells (CD45$^+$CD3$^-$CD207$^+$) increased at 4 h compared to NSK (Fig. 5d). A decreasing, though non-significant trend in the NK cell population frequency was observed over time (Fig. 5f). Regarding the macrophage population, M2-skewed macrophage (CD45$^+$CD14$^+$CD163$^+$) frequency increased after 4 h post-bite as well as their expression of activation markers (CD16/CD69) compared to NSK and 30 min (Fig. 5h–j). Frequencies of activated M2 macrophages and frequency of Langerhans cells positively correlated with clinically observed changes in bite size at 30 min (Supplementary Fig. 4a, b). In order to evaluate if changes in total leukocytes (CD45$^+$) were affecting the reported frequencies, we analyzed the changes of the innate cell population within total live cells. Here, similar results were observed (Supplementary Fig. 5). Unfortunately, we did not include specific neutrophil markers to our flow cytometry panels. Alternatively, when we gated on CD45$^+$CD11b$^+$CD14$^-$CD56$^-$CD117$^-$ cells comprised mostly of neutrophils, we detected a significant "neutrophil" accumulation at 4 h post-bite compared to NSK (Supplementary Fig. 6). The relative proportion of CD4$^+$ and CD8$^+$ T cells did not change over time. Taken together, the results suggest that immune cell infiltration and migration occurs in local

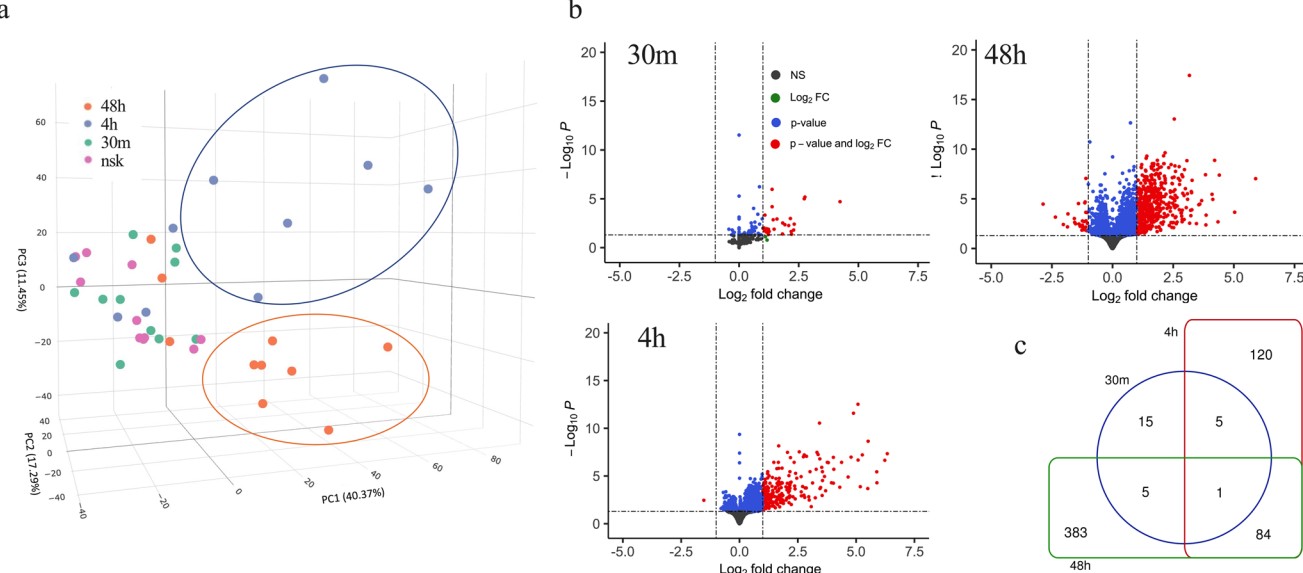

**Fig. 3 | Differential expression gene (DEG) induced by *Aedes aegypti* mosquito bites over time. a** 3D principal component analysis showing clustering of individual biopsies taken at 30 min (green dots), 4 h (blue dots), 48 h (orange dots) post-bite or normal skin (NSK; pink dots). Blue ellipse indicates clustering at 4 h, while orange ellipse at 48 h. **b** Volcano plot comparing DEG from skin biopsies post-bite with NSK (significant genes with −1 < log2 fold change (FC) > 1 and FDR < 0.05 are shown in red). FDR were accounted for by adjusting *p*-values using Benjamini-Hochberg multiple testing correction method (DESeq2 R-package). **c** Venn diagram summarizing the relationships between significantly upregulated genes over NSK (log2FC > 1 and FDR < 0.05) and time post-bite.

skin tissue within 4 h post-bite and is characterized by increased frequencies of DCs, M2 macrophages, and neutrophils, but decreased frequencies of NK cells.

### Changes in the T cell compartment are observable at 48 h after mosquito 'bite'

Changes in skin T cell populations were assessed using NSK and biopsies taken at 48 h post-bite. We found that the frequency of CD8+CD25+ and CD8+PD1+ T cells increased at 48 h post-bite, compared to the NSK (Fig. 6a). Frequencies of CD8+ T cells correlated inversely with the bite size at 48 h (Supplementary Fig. 4c). The frequency of effector memory cells re-expressing CD45RA (T$_{EMRA}$) CD8 T cells (CD8+CCR7-CD45RA+) decreased at 48 h (Fig. 6b). In the CD4+ T cell compartment, we observed a decrease in the frequency of T$_h$1/T$_h$17 (CD4+CCR4-CXCR3+) cells at 48 h compared to NSK and an increase in the frequency of T$_h$2/T$_h$17 (CD4+CCR4+CXCR3-) cells (Fig. 6c). The frequency of CD4+ effector memory T cells (T$_{EM}$) (CD4+CCR7-CD45RA-) increased at 48 h compared to NSK, while the frequency of T$_{EMRA}$ cells (CD4+CCR7-CD45RA+) decreased (Fig. 6d). Results were similar when analyzing the changes of T cell frequencies within all live cells (Supplementary Fig. 7). Interestingly, no changes were observed in the frequencies of skin-resident CD4 (CD4+CD103+CLA+) or CD8 (CD8+CD103+CLA+) T cells (Supplementary Fig. 8). At 48 h post-bite, these results show an increase in the frequency of activated CD8 T cells in addition to a shift from T$_h$1/T$_h$17 towards T$_h$2/T$_h$17 polarization of CD4 T cells.

### "Bitten" human skin cells stimulated by *Aedes aegypti* saliva produce significantly less pro-inflammatory cytokines

Next, we sought to assess if *Ae. aegypti* saliva impacted cytokine production by skin cells taken at 48 h post-bite. We measured pro- and anti-inflammatory cytokines released into the media after a stimulation assay with and without *Ae. aegypti* SGE. Both NSK and SGE-treated cells showed low secretion of cytokines in the media (Supplementary Fig. 9). However, we observed a significant reduction in the levels of Interleukin 2 (IL-2) and Interferon gamma (IFN-γ) in the cells treated with both SGE and PMA/Ionomycin compared to the cells treated with PMA/Ionomycin alone (Fig. 7, Supplementary Fig. 9). The levels of

other pro-inflammatory cytokines like TNF-α, IL-17A and IL-17F showed no significant differences (Fig. 7). Similarly, no differences in the regulatory cytokine IL-10 were observed (Fig. 7). These data suggest that SGE has a dampening effect on the production of pro-inflammatory cytokines IL-2 and IFN-γ under stimulatory conditions in 'bitten' human skin cells.

## Discussion

Arboviruses enter the host through the skin when the arthropod injects its saliva and other components into the dermis while taking a bloodmeal. Data from mice and macaques suggest that the unique dual presence of virus and saliva orchestrates an immunological cascade of events that influence the establishment and course of infection[12,16,17]. To translate these findings to humans, the current study assessed the dynamics of the cutaneous immune response to arthropod bites from a clinically relevant vector, the *Ae. aegypti* mosquito, in humans from arboviral-endemic areas with lifelong exposure to *Aedes* mosquitos (see Graphical Abstract Summary, Fig. 8). At 4 h post-bite, we observed upregulation of genes involved in neutrophil recruitment and increased frequencies of dendritic cells and M2 macrophages. As the immune response progressed over 2 days, we observed an increased frequency of activated, PD-1+CD8+ T cells, a polarization towards T$_h$2/T$_h$17 CD4+ T-cells, and upregulation of IL-10, IL-4, and IL-13 and IFN-γ related pathways. Interestingly, these human skin cells post-bite produced less pro-inflammatory cytokines when stimulated by *Ae. aegypti* SGE.

A mosquito 'bite' results in minor tissue damage by the proboscis, injection of mosquito saliva into the skin, and introduction of both mosquito and host microbiota into the skin. In this study, we did not parse out the separate impact of each process on the skin immune response. Alterations in skin immunity due to injury from the punch biopsy itself were mitigated by ensuring biopsy sites were at least 10 to 20 mm away from one another and were not taken directly at the site of the needle administration of subcutaneous lidocaine. The effect of local anesthetic was mitigated by the case-control design that each person's anesthetized, bitten skin was compared to anesthetized, unbitten skin, presumably negating any immediate cutaneous immune effects attributed to lidocaine, for which the literature is sparse.

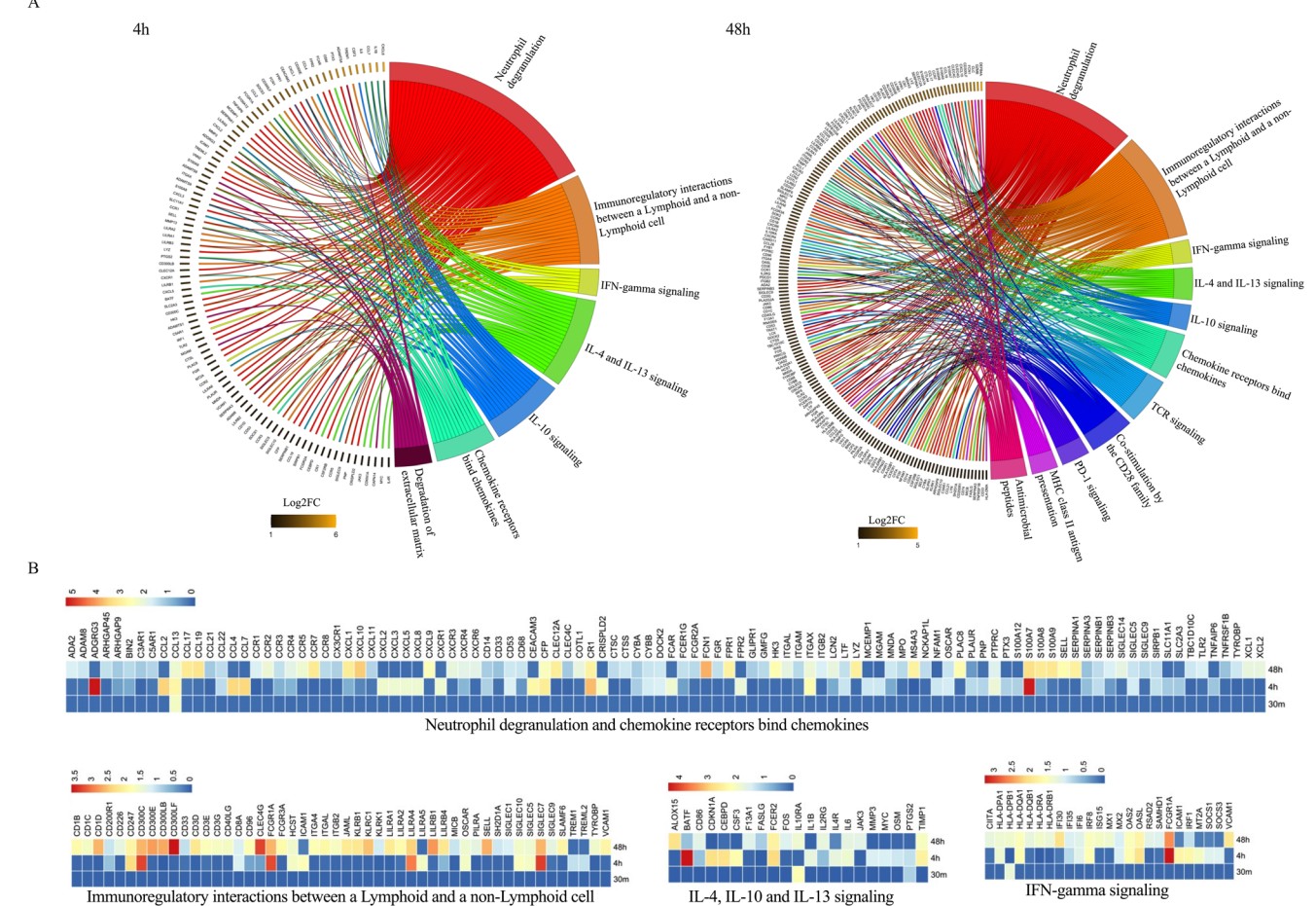

**Fig. 4 | Pathway analysis resulting from *Aedes aegypti* mosquito bites over time.**
**A** Chord plot shows significantly upregulated genes at 4 h and 48 h post-bite over NSK (left side of chord plot) and overrepresented pathways (right side of chord plot). Outer left ring shows genes and color range displays log2 fold change (FC) (left, key at bottom) or Reactome terms (outer right ring). Chords connect gene names with Reactome term pathways. **B** Heatmap (scaled to log2FC) displaying a list of DEGs based on the common pathways overrepresented at 4 and 48 h post-bite.

Mosquito bites cause a local cutaneous reaction with wheal and flare manifestation[20,21]. Concordantly, the bite size reaction peaked after 15 min post-bite in this endemic study population. Few changes were detected in gene expression or influx and activation of neutrophils, dendritic cells subsets, and macrophages at the 30 min post-bite time point. Upregulated genes were related to top-level processes such as signal transduction. We did not design this study to assess the role of pre-stored immune mediators such as histamine (and others) in the development of edema and their modulation in the immune response. Indeed, mosquito bites can activate cutaneous mast cells, which are prominent histamine-storing cells, and induce their degranulation. The release of histamine and other mediators leads to vasodilation, fluid accumulation at the bite site, and neutrophil influx, which were all ultimately observed here[20,21].

As the immune reaction progressed, the mosquito 'bite' induced degradation of the extracellular matrix (ECM) and a strong influx of dendritic cells and macrophages at 4 h post-bite, consistent with previous animal observations[22]. We observed several disintegrin and metalloprotease with thrombospondin type I motifs (ADAMTS) were upregulated, mostly at 4 h post-bite, that can be associated with ECM degradation. Similarly, granzyme B (GZMB) upregulated at 48 h; despite its effects on inducing inflammation it may also be involved in ECM degradation using as substrate fibronectin, vitronectin, laminin and others[23,24]. Saliva from blood-feeding arthropods contains anti-hemostatic proteins that prevent coagulation and promote vasodilation, leading to increased leukocyte trafficking[20,25]. These

processes – extracellular matrix degradation and vasodilation – can facilitate viral infection by disrupting vascular functions, as is the case of sialokinin, a mosquito salivary protein that hinders endothelial barrier integrity[26]. In addition, elevated leukocyte trafficking can facilitate viral replication by augmenting the proportion of susceptible cells to the local inoculation site[12,15,26]. Indeed, as most leukocyte subsets are permissive to infection by most arboviruses, they play a dual role in the infection process: both as replication targets/dissemination vehicles and as the first line of defense. The increased presence of activated M2 macrophages and dendritic cells at 4 h post-bite highlights the cellular trafficking role initiated by 'the bite' that can ultimately result in viral dissemination. The observed increase in CD163$^+$CD14$^+$ cells at 4 h is what we believe to be an initiation of the differentiation of monocytes and macrophages into M2-skewed macrophages given that full macrophage polarization does not occur as early as 4 h[27–29]. Upregulated chemokines such as CCL2 and CCL7, attracting mainly monocytes, DCs and T cells, may be participating in this recruitment[30]. M2-polarized macrophages produce anti-inflammatory cytokines, mediate wound healing processes, and support T$_h$2 responses[31]. Moreover, activation of M2 macrophages positively correlated with the bite size in our study. Upon stimulation, DCs upregulated CD69, preventing them from exiting the tissue and improving the likelihood of antigen internalization. Subsequent CD69 downregulation promotes Sphingosine-1-Phosphate (S1P1) leading to DC migration to the lymph nodes[32]. The CD69-S1P1 mechanism of cell retention in tissue

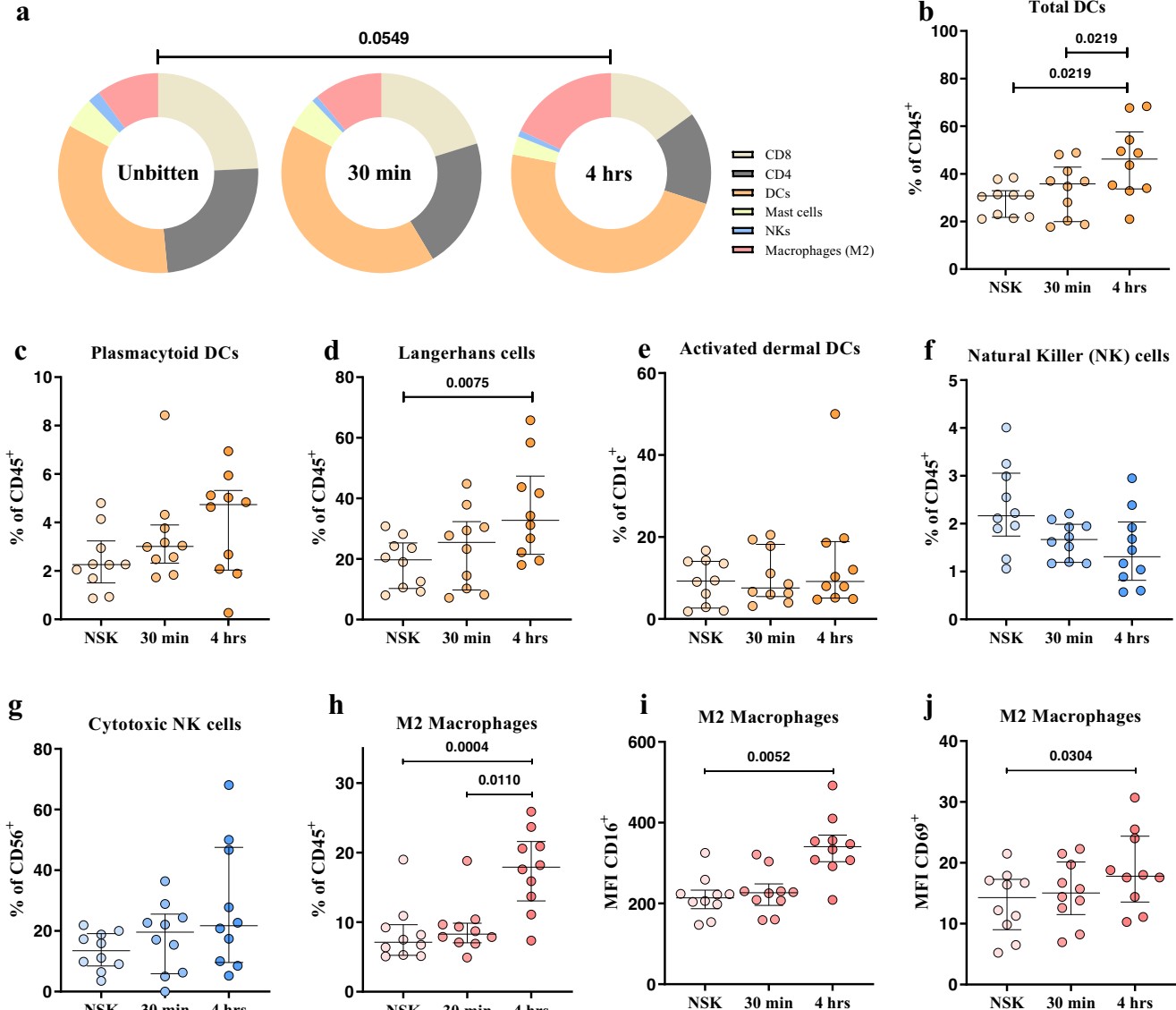

**Fig. 5 | Early innate immune responses in the skin are characterized by increased proportions of DCs and M2 macrophages, and a decreased proportion of NK cells at 4 h post-bite. a** Pie charts showing changes in the frequencies of skin immune cells during early innate immune response to mosquito bite at 30 min and 4 h after exposure. Frequencies are reported as percentages of total leukocytes (CD45[+]). **b** Total frequency of dendritic cells calculated as the sum of Langerhans (CD207[+]), dermal (CD1c[+]), and plasmacytoid (CD123[+]) DCs. **c, d** Frequency of plasmacytoid dendritic cells and Langerhans cells populations reported as percentage of total CD45[+] cells shows significant increase of the latter, at 4 h post

mosquito bites. **e** Frequency of activated dermal DCs (CD1c[+]CD69[+]) from the total dermal DCs population (CD1[+]). **f** Frequency of NK cells (CD56) reported as percentage of total CD45[+] cells. **g** Frequency of cytotoxic NK cells (CD56[+]CD16[+]) as percentage of total NK cells. **h–j** Frequency of M2 macrophages and MFI of their expressed activation markers CD16 and CD69. Statistical analyses were performed with Chi-square test, two tailed (**a**) and Friedman + Dunn's multiple comparisons test, two tailed with adjusted *p*-values reported (**b–j**). Bars indicate median and interquartile range. *N* = 10 individuals. Only *p*-values < 0.05 are reported. Source data are provided as a Source Data file.

is also well-described for resident T cells[33], whose role in saliva-mediated immunity is not yet known.

*Ae. aegypti* mosquito saliva has a clear impact on neutrophil influx and degranulation. Two recently identified *Ae. aegypti* salivary proteins (NeSt1 and AgBr1) attract and stimulate neutrophils at the 'bite site,' changing the immune microenvironment and leading to increased early ZIKV replication and enhanced pathogenesis[14,15]. Here, at 4 h post-bite, chemokines CXCL2, CXCL3, CXCL5, and especially CXCL8, plus the cytokine IL-1B were upregulated returning to levels comparable to NSK at 48 h post-bite. CXCL2-5 chemokines are responsible for attracting neutrophils and exhibiting angiogenic properties[34]. Pro-inflammatory responses through inflammasome activation and IL-1B production can trigger a sustained neutrophilic response[12]. Microbiota egested during infected sand fly bites

contributed to IL-1B production[35], but we were unable to assess the effects attributed to vector microbiota in this study. On the other hand, IL-20 was upregulated at 4 h post-bite. IL-20 is most active at epithelial sites downregulating IL-1B resulting in deactivation of neutrophil functions such as phagocytosis, granule exocytosis, and migration[36,37]. Increased gene expression for neutrophil recruitment and degranulation at 4 h post-bite in our Cambodian study population further supports the timing of neutrophil influx seen 3 h post-bite in bite-experienced C57BL/6 immunocompetent mice[12]. How the timing of neutrophil influx would differ in a bite-naïve population is a question that remains to be answered as none of the participants would be considered naïve. Moreover, studies of 'infected bites' would clarify the role of both 'neutrophil influx' and extracellular matrix degradation in facilitating pathogen establishment[12,26], but

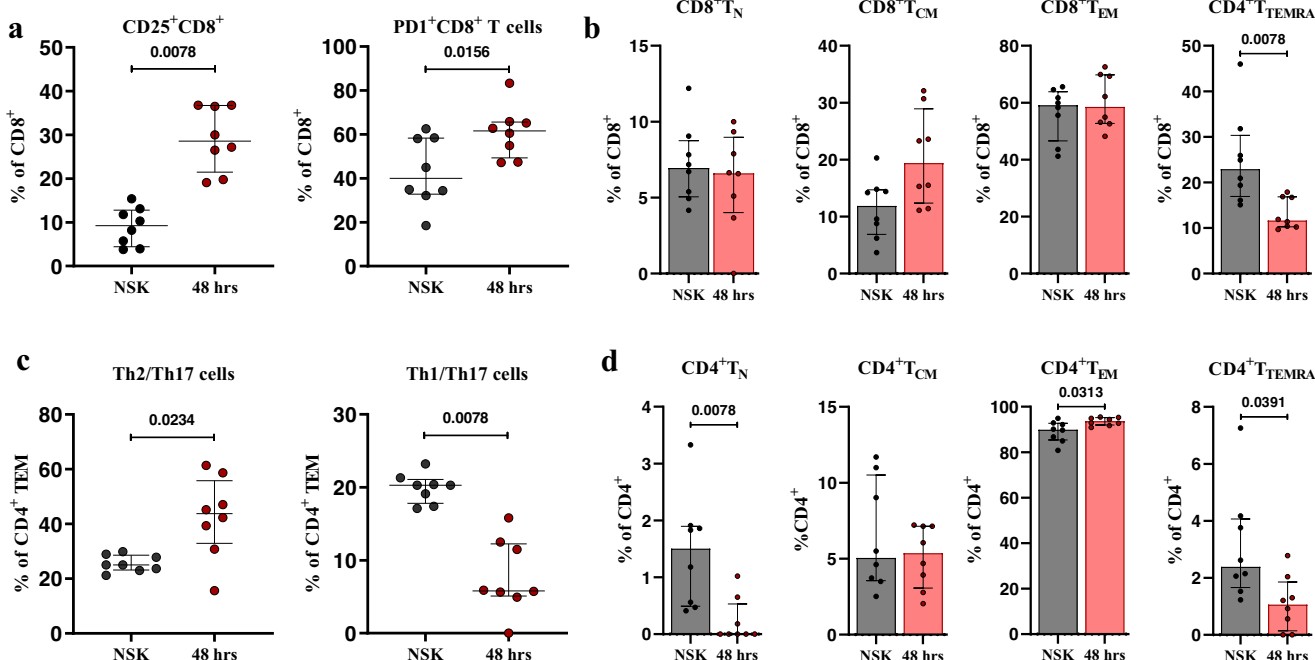

**Fig. 6 | Changes in the T cell compartment are observable at 48 h after mosquito exposure. CD8 T cells exhibit a high activation phenotype, while CD4 T effector memory cells shift from a $T_h1/T_h17$ phenotype to a $T_h2/T_h17$ phenotype. a** Increased frequency of CD25$^+$ and PD1$^+$ CD8 T cells at 48 h post mosquito exposure. **b** Frequencies of CD8 T cells naïve (CCR7$^+$CD45RA$^+$), central memory (CCR7$^+$CD45RA$^-$), effector memory (CCR7$^-$CD45RA$^-$), and TEMRA (CCR7$^-$CD54RA$^+$).

**c** Decreased $T_h1/T_h17$ and increased $T_h2/T_h17$ effector memory CD4 T cell compartment at 48 h. **d** CD4 T cells naïve (CCR7$^+$CD45RA$^+$), central memory (CCR7$^+$CD45RA$^-$), effector memory (CCR7$^-$CD45RA$^-$) and TEMRA (CCR7$^-$CD54RA$^+$) frequencies. Statistical analyses were performed with Wilcoxon signed-rank test two tailed (**a**–**d**). Bars indicate median and interquartile range. $N = 8$ individuals. Only $p$-values < 0.05 are reported. Source data are provided as a Source Data file.

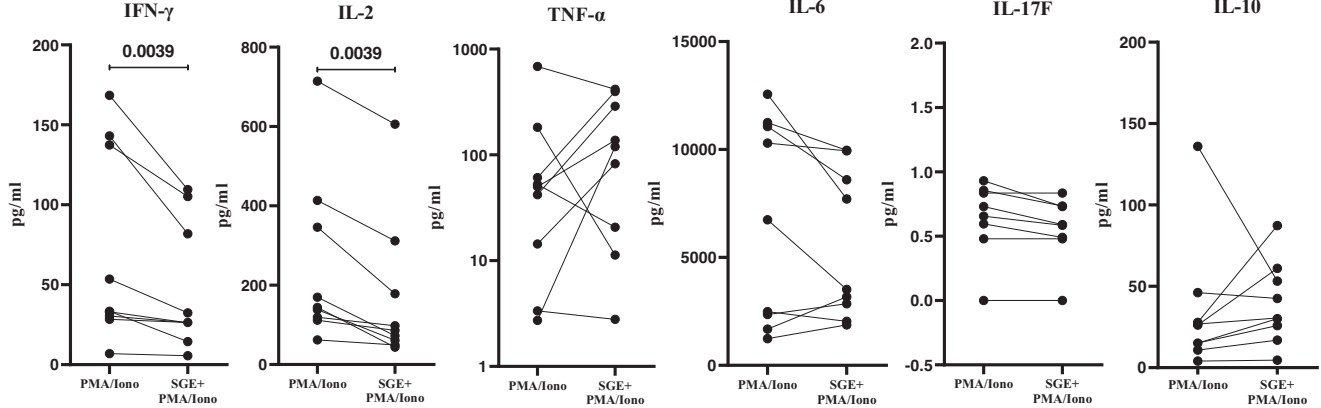

**Fig. 7 | Human skin cells obtained from *Ae. aegypti* bitten skin stimulated by *Ae. aegypti* salivary gland extract (SGE) produce significantly less pro-inflammatory cytokines (IL-2 and IFN-γ).** Cytokine production measured in skin cell culture supernatant after treatment with PMA/Ionomycin in the presence or absence of SGE. Skin dissociated cells were seeded on round-bottom 96-well plates

(50,000 cells/well) and treated with SGE (10 μg/mL) or PBS for 24 h. Cells were stimulated with PMA (0.1 μg/mL) and Ionomycin (1 μg/mL) or left unstimulated for the last 6 h of the culture. Statistical analysis was performed with Wilcoxon signed-rank test, two tailed. $N = 9$ individuals. Only $p$-values < 0.05 are reported. Source data are provided as a Source Data file.

currently approved human challenge models deliver DENV via needle inoculation, not via mosquito bite[38].

As the immune response transitioned from innate to adaptive, a strong T cell signature emerged at 4 and 48 h after the mosquito 'bite'. CD8$^+$ T cell activation was confirmed by increased expression of CD25, an early activation marker. Skin CD8$^+$ T cells also upregulate PD-1 expression, an immune-checkpoint receptor that is induced by yet can also inhibit T cell activation. PD-1 primarily inhibits effector function acquisition by T cells, results in the loss of sensitivity after T cell receptor signaling and modulation of cytotoxicity, and blocks TNF-α and IL-2 production along with T cell expansion[39,40]. Interestingly,

highly activated CD8$^+$ cells have been observed in skin from dengue shock syndrome patients[41,42]. Here at 4 and 48 h post-bite, CD4$^+$ T cells had a skewed phenotype to $T_h2$-like cells and IL-10, IL-13, and IL-4 related pathways were upregulated compared to NSK. CCL18, upregulated at 48 h post-bite, can be expressed by M2 macrophages and is linked to increased IL-4, IL-10, and IL-13, especially in the presence of histamine[43]. Indeed, initial animal experiments also pointed to a $T_h2$-mediated allergic inflammatory response following mosquito bites or inoculation with SGE[13]. In case of arbovirus infection, virus will replicate locally in the skin, and the local skin immune responses will direct the course of the systemic host immune response. At 48 h post-

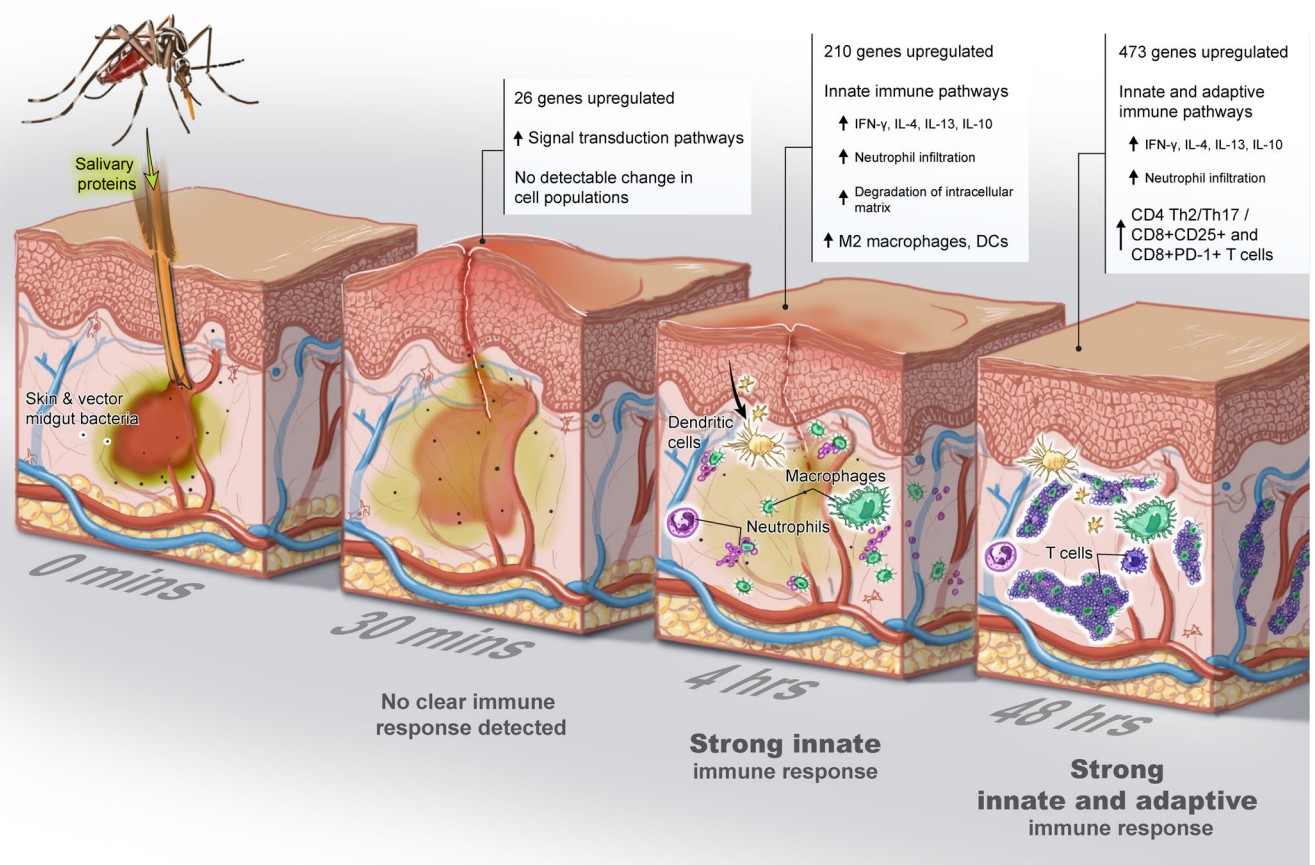

**Fig. 8 | The cutaneous immune response to *Aedes aegypti* mosquito 'bites' evolves over time.** Little activity in gene regulation and cell populations were noted at 30 min. At 4 h post-bite, a clear innate immune signature emerged followed by a strong adaptive response at 48 h.

bite, arboviruses will also have disseminated to draining lymph nodes[44]; hence, the contribution of the localized $T_h2$ polarization at this time point on disease outcome remains to be determined.

At 48 h post-bite, other upregulated genes such as CD28, ICOS, and CTLA-4 further suggest a regulatory role of the CD4+ compartment in controlling the local immune responses to *Ae. aegypti* saliva. $T_h2$ polarization is essential to avoid excessive collateral damage from an exacerbated pro-inflammatory response, especially in a situation where the skin immune cells are repeatedly exposed to the same type of insult, as is the case with individuals living in tropical areas of mosquito abundance. Blood DC antigen-2 (CLEC4C), upregulated at 48 h compared to NSK, is thought to signal through BDCA-2 to inhibit activation of the NF-κB pathway, inhibit production of type I IFNs, and block TRAIL-mediated cytotoxic activity[45]. This may represent a possible mechanism that differentiates chronically exposed individuals compared to naïve individuals.

To consider other potential mechanisms that immunomodulate the microenvironment, we noted increased expression at 48 h post-bite of ficolin-1 (FCN1), a complement lectin pathway protein that may bind and opsonize mosquito microbiota or salivary components, then activate the complement pathway[46]. Another C-type lectin receptor, LSECtin (CLEC4G, increased at 48 h compared to NSK), negatively regulates T cell-mediated immunity and may be also linked to apoptotic cell clearance by macrophages[47]. Of note, the upregulation of CXCL10 chemokine and its receptor CXCR3 are linked to their major roles of inflammatory reactions, T cell migration, and IFN-γ in the saliva-mediated process[48]. Besides chemokines, the top expressed molecule was an antimicrobial peptide hBD-2 (DEFB4A), produced by neutrophils and keratinocytes. DEFB4A is a marker of inflammation absent in normal skin and responsible

for recruitment of memory T cells, immature DCs, mast cells, and neutrophils[49,50].

In this study, the presence of mosquito saliva inhibited pro-inflammatory cytokine production by 'bitten' human skin cells after in vitro re-stimulation with SGE. One consideration is that we used SGE rather than mosquito saliva as forced salivation did not provide sufficient material, but our preparations minimize the cellular components of the salivary glands. The cytokine-dampening effect may partly result from the interaction between salivary components and key checkpoints on different signaling pathways. One example includes the salivary protein LTRIN known to interfere with the NF-κB pathway by binding to LTβR, thus hindering its dimerization and leading to decreased production of inflammatory cytokines[11]. On the other hand, skin immune tolerance to constant mosquito exposure may also explain the effect of SGE to reduce cytokine production as seen here in our bite-experienced participants[12,19,51,52].

Our results refine and extend the knowledge obtained from mouse models of mosquito biting. For example, in mouse models, increased frequencies of dendritic cells, total macrophages and neutrophils have been detected after mosquito bites[12,21,53]. In our human participants, we also observe an increase in the frequency of these cells, indicating that the mouse models of mosquito biting recapitulate the early innate events in human skin. However, we expand these data by showing that the observed increased frequencies are due to increases in M2-like macrophages, which are activated, and increases in Langerhans cells at 4 h post-bite in our human participants. Additionally, we substantially increased our knowledge on human T cell responses to mosquito saliva by observing CD8 T cells exhibiting a high activation phenotype, while CD4 T effector memory cells shift from a $T_h1/T_h17$ phenotype to a $T_h2/T_h17$ phenotype 48 h post-bite.

Saliva or salivary proteins from ticks and sand flies have shown ability to inhibit proliferation of both murine and human cells[54,55]. We here confirm the inhibitory effects of SGE to skin cells recruited during a mosquito bite. Nevertheless, as previously showed in individuals from an endemic area of leishmaniasis[56], the immune responses to saliva of arthropods also triggers a pro-inflammatory response highlighted by the upregulation of pathways related to interferon genes. This mix of anti- and pro-inflammatory responses need to be further evaluated in animal models of mosquito bite exposure as well as clinical field settings.

With the current rise of vector-borne diseases around the world, future clinical development of vector-targeted strategies for both known and unknow vector-borne diseases will require a fundamental understanding of how vector saliva modulates the cutaneous immune response of humans. Translational studies, such as the one presented here, significantly expand our current knowledge on human skin immunity and the immune response orchestrated by a 'mosquito bite'. The genes, cell types, and processes of human cutaneous immunity identified here can be leveraged in the development of vector-targeted vaccine candidates or novel therapeutics such as those targeting the reversibility of functional macrophage polarization or chemokine blockades that may have application in other inflammatory disorders[57]. Such new pathogen-agnostic strategies have shown promising results in recent years from mosquito saliva-based vaccinations to topical applications like imiquimod post-bite that activated skin macrophages in order to limit pan-viral activity[58–61]. Building on these findings, the next generation of countermeasures against vector-borne disease will require nuanced and comprehensive understanding of skin immunity to pathogen and vector saliva alike.

## Methods

### Ethics

This study was approved by the National Institutes of Health Institutional Review Board and the National Ethics Committee on Human Research in Cambodia.

### Study enrollment and procedures

From October 2020 to January 2021, we recruited and enrolled thirty Cambodian adults at Kampong Speu District Referral Hospital. Eligible participants were healthy adults between the ages of 18 and 45 years old who agreed not to use topical or oral antihistamines or steroids for the duration of the study. Participants were excluded if they were pregnant or reported a history of severe allergic reaction to mosquito or other insect bites, use of medications that affect blood clotting, or history of significant scarring such as keloid formation (full criteria found on https://clinicaltrials.gov/ct2/show/NCT04350905). All participants provided written informed consent before enrollment, and the study was conducted in accordance with the provisions of the Declaration of Helsinki and Good Clinical Practice guidelines. Study protocol is available upon request.

On Day 0, five starved uninfected *Ae. aegypti* female mosquitos were selected from a mosquito colony dedicated to human feeding studies in the Malaria and Vector Research Laboratory at the National Center for Parasitology, Entomology, and Malaria Control. Mosquitos fed through the disposable mesh of a feeding device placed on participants' ventral forearm skin for 10 min. Redness and swelling in millimeters were assessed immediately, 15 min, 30 min, 4 h, and 48 h after feeding. Redness and swelling were measured in millimeters at 15 min, 30 min, and 4 h.

A total of five 3-mm skin punch biopsies were performed on each individual: one biopsy from bitten skin per time point (30 min, 4 h, and about 48 h after feeding was completed) for a total of three 'bitten' biopsies, and two from NSK collected at baseline. Biopsy sites were selected based on the 3 bite sites with the largest diameters at 30 min after feeding. We tattooed the area surrounding those 3 bite sites with

a permanent marker and randomly picked one of the three distinct sites for 30-min, 4-h, and 48-h biopsies. Less than 1 cc of 0.2% lidocaine without epinephrine was delivered via 30-G needle at each chosen 'bite' at the time of biopsy and 3-mm biopsy was performed on the 'bite' adjacent to needle puncture. For feasibility of skin sample analysis, each participant was assigned to 1 of 3 technical modality cohorts: (1) immunohistochemistry, (2) RNASeq, and (3) flow cytometry with ten people in each cohort. Additionally, blood was collected for serological and cellular analyses at baseline and 14 days after feeding. Patients were compensated per local IRB norms.

### *Aedes aegypti* SGE preparation

Salivary glands from *Ae. aegypti* were dissected and collected in 200 µl of PBS 1x (1 pair/1 µl of PBS 1x) in low protein binding centrifuge tubes. Homogenization was done in an Ultrasonic SONIFIER® (BRANSON) for 30 s on each side of the tube. The homogenates were then centrifuged at $25,000 \times g$ at 4 °C for 3 min, and the supernatant was collected and quantified using Pierce™ BCA Protein Assay Kit (ThermoFisher Scientific, Cat# 23225) in an xMark™ Microplate Spectrophotometer (BIO-RAD).

### *Aedes aegypti* salivary IgG enzyme-linked immunosorbent assays

*Ae. aegypti* SGE was diluted to a concentration of 2 µg/µl in carbonate-bicarbonate buffer (Millipore-Sigma) and used to coat 96-well Immulon 4x HB plates (Nunc) overnight at 4 °C. Plates were blocked in a solution of TBS 4% bovine serum albumin and 0.025% Tween 20 (Millipore-Sigma). Each plate included sera (1:200 dilution) from all time points, a negative control (non-reactive human sera to SGE), an internal control (a pool of positive sera to SGE with OD set to 0.25 at 450 nm), and blank wells. The controls were chosen accordingly from a stored bank of human sera, well-characterized by reactivity to the saliva of various vectors, at the Laboratory of Malaria and Vector Research. Plate-to-plate normalization was done by multiplication of all OD values by a correcting factor to achieve the preset internal control OD 0.25 at 450 nm. Secondary antibody [goat anti-human IgG alkaline phosphatase (1:10000 in blocking solution; Millipore-Sigma)] was used. A solution of p-nitrophenylphosphate (Millipore-Sigma) was used as substrate and OD measured at 450 nm in arbitrary enzyme-linked immunosorbent assay (ELISA) units and reported in arbitrary ELISA units after correction stated above.

### Dengue IgG enzyme-linked immunosorbent assays

The PanBio® Dengue Indirect IgG ELISA (product code 01PE30) was performed according to the manufacturer's instructions. Two microtiter plates were supplied, one containing stabilized dengue-1 through dengue-4 (antigen plate) and the other containing anti-human IgG bound to separate wells (assay plate). Peroxidase-labeled anti-dengue virus monoclonal antibody (125 µl/well) was added to the antigen plate to solubilize the antigens and form antibody-antigen complexes. Concurrently, 100 µl of patient serum were diluted 1:100 in the provided diluent and added to each well of the assay plate containing bound anti-human IgG, and human IgG in the patient's serum was captured. The plates were incubated for 1 h at room temperature (antigen plate) or 37 °C (assay plate); then the assay plate was washed, and 100 µl of antibody-antigen complexes per well were transferred from the antigen plate to the assay plate. These complexes were captured by dengue virus-specific IgG during incubation for 1 h at 37 °C. The plate was washed, and bound complexes were visualized through the addition of 100 µl of tetramethylbenzidine substrate per well. After 10 min, the reaction was stopped by the addition of 100 µl of 1 M phosphoric acid per well, and the strips were read at 450 nm with a microtiter plate reader. Positivity was determined by comparison to the IgG reference sera provided (cutoff calibrators). A positive sample was defined as having a

sample/calibrator absorbance ratio of ≥1.0, and a negative sample was defined as having a ratio of <1.0.

## Formalin preservation and H&E staining of skin biopsies
Biopsies were placed in 7-ml prefilled 10% formalin vials (Azer Scientific) and stored at 4 °C until paraffin embedded. Unstained slides were cut from paraffin blocks and deparaffinized in xylene, then hydrated through graded alcohols up to water, placed in Carazzi's hematoxylin, washed in tap water, and put in one change 95% ethanol. Next, they were placed in eosin-phloxine solution and run through graded alcohols to xylene. After xylene, the stained slides were covered with a cover slip and mounted using Permount Mounting Media (Fisher Scientific).

## RNA extraction, RNA-seq library preparation, sequencing, and analysis
Biopsies were stored in DNA/RNA Shield (Zymo research) and kept at 4 °C until processing. Skin was homogenized using 3-mm zirconium beads (Biolink laboratories) and a Magnalyser bead beater (Roche Diagnostics®). After homogenization total RNA was loaded at Innu-Pure® C16 touch (AnalitikJena) for magnetic separation of total RNA using an InnuPREP® RNA Kit–IPC16 (AnalitikJena). Quality of extractions was verified on a 4200 TapeStation (Agilent) and all samples had RIN numbers above seven. RNA sequencing was performed by Novogene. Briefly, mRNA was enriched by Poly (A) capture and RNA libraries were prepared using the NEBNext Ultra II Directional RNA Library Prep Kit for Illumina (New England Biolabs) and sequenced on a NovaSeq 6000 (Illumina) generating paired end reads at 150 bp length. After removing low quality reads (Trimommatic 0.39)[62], mapping to the human genome (hg38; GCF_000001405.39) and gene-level quantification were generated by RSEM 1.3.2[63], Bowtie 2-2.4.2[64] and SAMtools 1.11[65]. Normalization and differential gene expression (DGE) analysis were performed using DESeq2 1.34.0[66], employing the variance stabilizing transformed function (VSN 3.62.0)[67] to the count matrix that was designed to account for paired samples enabling expression analyses with normal skin samples from the same subject as reference. Approximate Posterior Estimation for GLM (Apeglm 1.16.0) method was used to calculate fold change (FC) of DEGs[68]. DEGs were considered significant if apeglm-estimated shrunken log2FC was <1 or > 1 and with a false discovery rate <0.05 (Benjamini−Hochberg multiple testing correction method)[69]. PlotPCA3D 0.0.1(twbattaglia/btools) was used to generate tri-dimensional principal component analysis plots on DESeq2. EnhancedVolcano 1.12.0 was used to generate volcano plots[70]. GOplot 1.0.[71] was used to generate chord plots using data obtained from packages clusterProfiler 4.2.0[72] and ReactomePA 1.38.0[73]. Pathways with a Benjamini−Hochberg corrected *p*-value < 0.05 and were considered as enriched, selected pathways were mapped on chord plots[69]. Heatmaps were prepared using packages pheatmap 1.0.12.[74] and dendsort 0.3.4[75]. All DEG analyses were performed using R 4.1.0[76].

## Skin biopsy dissociation
Biopsies were transported in RPMI media (Sigma-Aldrich) at 4 °C. Whole skin dissociation kit (Miltenyi®) was used to perform the biopsies dissociation. An enzyme mix consisting of enzyme D, enzyme A, and buffer L was prepared according to the manufacturer's specifications. Single punch biopsies were incubated in 400 µl of dissociation mix, at 37 °C −5% CO$_2$ overnight. Following incubation, each sample mix was transferred to a Falcon round-bottom tube with cell-strainer cap (40−75 µm). Using a rubber pestle and the cell-strainer cap as a mortar, biopsies were gently macerated to increase cell recovery yield. 500 µl of cold RMPI was used to rinse caps. Samples were then centrifuged at 428 × *g*, 4 °C for 10 min and the pellet resuspended in 500 µl of cold RPMI. Dissociated skin cells were counted using a pre-programmed setting in a Countess™ II FL Automated Cell Counter.

## Flow cytometry
Dissociated cells were seeded on round-bottom 96-well plates and washed with 200 µl of ThermoFisher's PBS buffer 1x (428 x g, 4 °C for 5 min). Twenty µl of Biolegend Zombie Aqua fixable viability stain (1:500) was added and incubated for 20 min in the dark at 4 °C. Excess viability stain was washed away with 200 µl of PBS/BSA/EDTA buffer (428 x *g*, 4 °C for 5 min) and the pellet resuspended and incubated for 10 min at 4 °C in 4 µl of Biolegend's FcR blocking antibody (1:10). Cells were then incubated for 30 min at 4 °C with surface staining antibody master mix. After washing free unbound antibody with 200 µl of PBS (428 x *g*, 4 °C for 5 min), cells were fixed with 100 µl of Biolegend's fixation buffer 1x for 20 min at 4 °C in the dark. Following fixation, cells were permeabilized and washed with 100 µl of Biolegend perm-wash solution 1 x (428 x *g*, 4 °C for 5 min) and the pellet resuspended and incubated for 30 min at 4 °C in intracellular staining antibody master mix (antibody dilution 1/100). Stained cells were placed in 200 µl of PBS/BSA/EDTA 1x and visualized on a BD Biosciences FACS Aria II using FACS Diva software. Analysis was performed using BD Biosciences FlowJo software. Antibody panels are described in supplemental material (Supplementary Table 2).

## In vitro SGE stimulation and multiplex cytokine screening
Skin-dissociated cells were seeded on round-bottom 96-well plates (50,000 cells/well) and treated with SGE (10 µg/mL) or PBS for 24 h. Cells were stimulated with PMA (0.1 µg/mL) and Ionomycin (1 µg/mL) or left unstimulated for the last 6 h of the culture. The supernatants were collected and stored at −20 °C. Cytokines in the cell culture supernatants were quantified using Human Th Cytokine Panel (12-plex) (Cat. 741027, BioLegend) according to the manufacturer's instructions. All samples in the experiment were measured in duplicate. Briefly, undiluted supernatants were incubated with beads coated with capture antibodies specific for IL-2, 4, 5, 6, 9, 10, 13, 17 A, 17 F, 22, IFN-γ and TNF-α for 2 h at room temperature on shaker. After incubation, beads were washed and incubated with biotin-labeled detection antibodies for 1 h, followed by a final incubation with streptavidin-PE for 30 min at room temperature on shaker. Beads were washed and re-suspended with washing buffer. Beads were analyzed by flow cytometry using a FACS Canto cytometer. Analysis was performed using the LEGENDplex analysis software v8.0, which distinguishes between the 12 different analytes on basis of bead size and internal dye.

## Statistics and reproducibility
Given no prior saliva skin immunity studies published in humans, feasibility and sample sizes are adapted from literature concerning neoplastic or autoimmune disorders of the skin versus healthy comparator skin. No data were excluded from the analyses, but TCR sequencing analyses are ongoing for an additional 10 patients. The experiments were not randomized and the investigators were not blinded to allocation during experiments and outcome assessment.

Flow cytometry and multiplex cytokine screening data were analyzed with GraphPad's Prism 9 software, using the non-parametric-based tests, Friedman + Dunn's multiple comparisons test, Wilcoxon signed-rank and Chi-square test. Correlation analysis was performed with the Pearson's correlation coefficient. Bars indicate median and interquartile range. Statistical analysis of RNAseq data is described above.

## Reporting summary
Further information on research design is available in the Nature Portfolio Reporting Summary linked to this article.

# Data availability
The Illumina raw reads (fastq files) were loaded at the NCBI Sequence Read Archive with accession numbers SRR18215008 to SRR18215047 and deposited in GenBank under bioproject accession code

PRJNA812009. All data generated or analyzed during this study are included in this published article (and its supplementary information files). Clinical trial data is available at https://clinicaltrials.gov/ct2/show/NCT04350905. Source data are provided with this paper.

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

## Acknowledgements

We would like to thank the study participants in Kampong Speu, Cambodia. We would also like to acknowledge Noemi Kedei and Maria Hernandez for their assistance in histology and staining, Jon Fintzi for statistical review of the clinical protocol, and Rose Perry-Gottschalk for her medical illustrations. This work was funded by: Division of Intramural Research at National Institute of Allergy and Infectious Diseases at the National Institutes of Health #1ZIAAI001066-12: J.M., F.O.; Howard Hughes Medical Institute-Wellcome #208710/Z/17/Z: T.C.; French National Research Agency #ANR-**17**-CE15-00029: D.M.; Calmette-Yersin PhD Scholarship: D.G.

## Author contributions

Conceptualization: F.O., J.E.M., T.C., J.G.V. Methodology: D.G., H.T.M.V., S.N., S.C., S.L., J.A.B., F.O., J.E.M. Investigation: D.G., H.T.M.V., J.A.B., S.N., S.C., S.L., A.R.P., J.G.V., F.O., J.E.M. Visualization: D.G., F.O., J.A.B. Funding acquisition: F.O., J.E.M. Study enrollment and clinical procedures: C.L., S.L., R.S, S.S., S.N., J.E.M, H.K, R.L. Project administration: C.L., T.C., F.O., J.E.M. Supervision: T.C., D.M, F.O., J.E.M., C.L, R.L., H.R, H.K. Writing original draft: D.G., H.V., T.C., F.O., J.A.B., J.E.M. Writing review and editing: T.C., D.M., C.L., F.O., J.A.B., J.E.M. All authors contributed to the article and approved the submitted version.

## Funding

## Competing interests

The authors declare no competing interests.

## Additional information

David Guerrero[1,7], Hoa Thi My Vo[1,7], Chanthap Lon[2,7], Jennifer A. Bohl[3], Sreynik Nhik[2], Sophana Chea[2], Somnang Man[2,4], Sokunthea Sreng[2,4], Andrea R. Pacheco[2], Sokna Ly[2], Rathanak Sath[2], Sokchea Lay[1], Dorothée Missé [5], Rekol Huy[4], Rithea Leang[4], Hok Kry[6], Jesus G. Valenzuela [3], Fabiano Oliveira [3,8], Tineke Cantaert [1,8] & Jessica E. Manning [2,3,8] ✉

[1]Institut Pasteur du Cambodge, Pasteur Network, Phnom Penh, Cambodia. [2]International Center of Excellence in Research, National Institute of Allergy and Infectious Diseases, National Institutes of Health, Phnom Penh, Cambodia. [3]Laboratory of Malaria and Vector Research, National Institute of Allergy and Infectious Diseases, National Institutes of Health, Rockville, MD, USA. [4]National Center of Parasitology, Entomology, and Malaria Control, Phnom Penh, Cambodia. [5]MIVEGEC, Univ. Montpellier, IRD, CNRS, 34000 Montpellier, France. [6]Kampong Speu Provincial District, Ministry of Health, Phnom Penh, Cambodia. [7]These authors contributed equally: David Guerrero, Hoa Thi My Vo, Chanthap Lon. [8]These authors jointly supervised this work: Fabiano Oliveira, Tineke Cantaert, Jessica E. Manning. ✉e-mail: jessica.manning@nih.gov

