## [Peer Review File · Nature Communications]

Evaluation of cutaneous immune response in a controlled human in vivo model of mosquito bitesREVIEWER COMMENTS

Reviewer #1 (Remarks to the Author):

GENERAL COMMENTS

Guerrero and colleagues assessed the cutaneous innate and adaptive immune responses via controlled *Aedes aegypti* feedings in humans living in an *Aedes*-endemic country during a First in Human study. Specifically, they compared the immune responses before and after *Ae. aegypti* bites in skin biopsies from 30 healthy Cambodian individuals living in an area where *Ae. Aegypti* mosquitoes are prevalent; all participants had detectable IgG to *Ae. aegypti* salivary gland extract (SGE) prior to mosquito feeding. The team observed induction of neutrophil degranulation and recruitment of skin-resident dendritic cells and M2-macrophages, followed by T cell priming and regulatory pathways along with a shift to a Th2-driven response and CD8+ T cell activation.

Prior animal/in vitro studies have shown that arthropod inoculation of vector saliva, and/or concomitant blood-feeding by an arthropod, can immunomodulate the host antiviral response in the skin and periphery, with dampening of the proinflammatory responses, recruitment of infection-susceptible cells to the bite site, stimulation of autophagy activation, and an induction of neutrophil infiltration.

While the study is largely descriptive, the clinical study and assays are well-executed and the human data are unique and important. The Discussion does an admirable job of synthesizing the array of assays and immune responses to generate a clear narrative that weaves together the sequence and interacting aspects of host responses that develop after mosquito biting. The findings are relevant to better understanding transmission of important vector borne pathogens and can be useful to conceive potential interventions.

While the authors highlight findings that are consistent with earlier preclinical studies, I would have liked to learn where the data here diverge from earlier animal or in vitro studies, if at all. Do the data diverge at all? The fact that the population has pre-existing *Aedes* antibodies and apparently extensive exposure to *Aedes* makes one suspect that some responses here would be modulated from what one might observe in animal or in vitro studies (or in populations with minimal exposure to *Aedes*). In that regard, the evidence that salivary gland extract inhibits specific pro-inflammatory responses of mosquito-bitten skin cells fits with this general idea.

Specific issues

1. How were biopsy sites selected? Non-random selection (eg, selecting the largest lesions) could bias the pattern of the responses seen.
2. Please standardize the colors when the same pathways are seen in the two panels of Fig 3D—it is confusing when you show the same pathways as different colors in upper and lower panels. Conversely, different pathways should NOT share the same color in the two panels.
3. Fig 4B—panel is labelled as “Total DC” but y axis indicates these are measured as “% of DC”. Please clarify meaning here of the measurement indicated by y axis—as written, “total DC” should consistently be measured as 100% for “% of DC”, no? One value is >100%-- how does that happen? What is the denominator here?

Reviewer #2 (Remarks to the Author):

The main strength of this report is its comprehensive nature and the use of relevant human skin sourced from individuals who live in regions of the world that are affected by mosquito-borne virus disease. The report contains a detailed description of these human skin responses to mosquito biting, from a clinical dermatology, histological, whole transcriptome analysis and cellular response perspective.

Corrections required

“First in human evaluation” is not accurate, as there have been previous longitudinal studies on human skin inflammatory responses to mosquito biting e.g. a recent example; <https://pubmed.ncbi.nlm.nih.gov/30353912/> This sentence should be modified to clarify and the ref should be added to e.g. line 67.

Line 55 -56. There is little evidence to suggest that mosquito biting leads to immune suppression within the time frame relevant for responses to arbovirus infection (lymphotoxin responses take many days to weeks) . The overwhelming majority of studies in mice show that the host response to mosquito biting is a pro-inflammatory event that does not suppress innate immune responses to virus. This should be rephrased.

Ref 12 and 32 appear to be duplicates.

The paper refers to “extended data” which I believe are supplementary files? These should be re-worded accordingly. Also, they are out of order (e.g. Extended data 3 is referenced in the text before extended data 2).

Figure 2

Excellent description and summary of human population, with post exposure to dengue and Aedes salivary proteins measured. However, it is not clear in figure 1 legend how bites were biopsied. As described, it gives the impression that an individual bite site was biopsied three times (30 mins, 4 hours and 48 hours). If this is the case, then the act of biopsy would alter inflammation at the site for later time points. If the authors mean that separate bite sites were biopsied for different time points, then this should be made clear.

Figure 3

The gene expression analyses is well done, correctly powered and highlights interesting features in host response. This analysis suggests that innate immune gene induction is present by 4 hours (eg cxcl8, and other neutrophil associated functions) and that genes associated with adaptive immune response and wound healing are present by 48 hours. This is perhaps not completely unexpected but importantly does suggest findings from mouse models, that have similar early induction of neutrophil responses, are broadly applicable to human infection. That mouse models of mosquito biting recapitulate these seen here in human skin is very helpful for the research community to know and should be more explicitly stated.

Line 182 – this is wrong figure – do you mean Supplementary figure 6?

Figure 4.

- It is not clear why there is no description of neutrophils numbers. Especially as the transcriptome analysis reveal neutrophils associated pathways as the most upregulated. This is a deficiency that cannot be easily corrected as fresh human skin samples would be

required. Especially for earlier time point, if histological sections exist, these should be stained for a neutrophil specific marker and numbers of neutrophils quantified in an unbiased fashion.

- It is not clear what 'frequency' refers to in each figure panel. Is this frequency of cell to total live cells, or of 'parent', or of CD45+ve cells? As these are whole tissues analyses, frequency is most accurately made in reference to per live cell (e.g. Langerhan cells per live cell), as increases in total cd45+ cell count could affect reported frequencies. Please clarify and re-analyse as required. This may explain why there is a reported apparent increase in Langerhans cells. This is unexpected as these cells tend to become rapidly depleted in skin following inflammatory stimulus and then are only replaced in the medium term (e.g. days to weeks in mouse models). This should be reflected in the description of the results.
- There is also an apparent increase in M2 macrophages. Is this likely? (no). The authors should report and discuss this carefully. As time required for M2 macrophage differentiation is far longer than 4 hours.

Figure 5

- As for figure 4, it is not clear what frequency is being referred to. As whole tissue analysis, this should be a frequency compared to number of live cells (not frequency of parent).
- There is no evidence that these CD8 T cells are exhausted. This reference should be removed and cells simply referred to as "activated".
- Most viruses have successfully disseminated via lymph, and infected tissues remote from the bite site, before this 48 hour time point (albeit as informed by mouse studies e.g. see <https://pubmed.ncbi.nlm.nih.gov/21147918/>). Therefore, any response in the skin at this later time point may have questionable consequence for defining outcome to infection. This should be stated clearly and conclusions made changed accordingly.

Figure 6

- This is an interesting figure, but it lacks an important control and some key experimental details. It would be most informative to compare data from mosquito-bitten skin to normal resting skin, ideally from mosquito bite unexperienced individuals, or at least from sites that are unlikely to have experienced biting. That way it would be possible to conclude whether mosquito biting influences capability of tissue to respond to further activation with saliva or SGE.
- The figure also lacks experimental detail
 - o time of stimulation with PMA or SGE?
 - o The amount of SGE used?
 - o How SGE was derived?
- It is not clear to what degree SGE replicates exposure fo cells to saliva from mosquito. For example, SGE will contain many cellular components that are not present in mosquito saliva. It is perhaps not surprising that a cellular homogenate has a detrimental effect on cultured cells. Quantification of indicative cell survival analysis should be done to show that cells are not dying, which would explain decreased cytokine expression.
- Due to these issues with SGE, this experiment would be best done using saliva obtained by forced salivation from mosquitoes. If not possible, this limitation should be discussed.

Discussion

- The first paragraph contains much needed methodological detail. This should be removed and inserted into the results section to clarify approach.
- Much of the language is too generalised and not specific enough. E.g. "innate immune cells" entry at 30 minutes is not accurate (line 285-286). The analysis of the most relevant innate immune cell type (neutrophils) was not done (e.g. mouse studies show rapid entry of

neutrophils in response to a variety of arthropod biting e.g.
<https://pubmed.ncbi.nlm.nih.gov/18703742/> <https://pubmed.ncbi.nlm.nih.gov/27332734/> .
The cell types studies should be specified.

-

Reviewer #3 (Remarks to the Author):

Focus on the skin as earliest frontline of host/pathogen interaction is interesting as it orchestrates downstream immune reactivity. The in principle descriptive data provide an comprehensive overview of genes, cell types and pathways in response to Aedes mosquito saliva in previously exposed volunteers over time. The paper is well written.

Major comments

The context of the study is placed in the background of Aedes transmitted viral diseases. The study design is limited to the responsiveness to uninfected mosquitoes in dengue-endemic participants. Comparison with infectious bites would have substantially increased relevance but obvious ethical constraints will prohibit such a study. An alternative design would have been studies in non-exposed Aedes volunteers (e.g. from non-endemic areas) complemented with infected bites using the existing controlled human dengue infection models. This should be mentioned and discussed.

Intensity and frequency of previous mosquito bites will have a great influence and likely affect responses. Included adult participants do have a history of Aedes exposure and Aedes aegypti salivary IgG enzyme-linked immunosorbent assays is used. Conclusion is made that there are no intra-group variations. However, there is no reference to the value and/or validation of this assay in relation to intensity or duration of mosquito exposure. Age, housing and geo-location are established parameters affecting exposure and biting rates. The authors indicate that the findings can be “leveraged to develop novel therapeutics and vector-targeted vaccine”. Please elaborate and make suggestions.

125: “These data indicate that time post-bite rather than inter-individual differences drive unique patterns in gene expression”. I disagree and would limit the claim to the conclusion that inter-individual differences do contribute to bite-driven patterns.

Fig 4+5: Legend: Please clarify “% of immune cells” compared to what? Provide the absolute numbers of the 100% values in the legend.

Fig 4: The 4hrs data indeed show significant changes compared to 30' or NSK time points at group level but range is often large. When analyzing individual data in the different figures, do volunteers cluster in relative strong and weak responders when comparing this different cell-types? Relation with dengue or mosquito previous exposure?

Fig 6: Cell number and composition of the stimulation assays is not provided. This complicates and limits interpretation of the data. Moreover, only 7/10 volunteers are presented. In vitro responses to Dengue antigens could have been informative but were not included.

333: Reference is made to injected microbiota in other diseases while saliva is considered a major trigger (319). Since the mosquitoes used were not sterile, could injected bacteria play a role in the generated data and responses of individual volunteers? Please comment

Minor comments

Is there a particular rationale for the selected time points e.g. hypersensitivity?

Response to Reviewers

Reviewer #1 (Remarks to the Author):

GENERAL COMMENTS

Guerrero and colleagues assessed the cutaneous innate and adaptive immune responses via controlled *Aedes aegypti* feedings in humans living in an *Aedes*-endemic country during a First in Human study. Specifically, they compared the immune responses before and after *Ae. aegypti* bites in skin biopsies from 30 healthy Cambodian individuals living in an area where *Ae. Aegypti* mosquitoes are prevalent; all participants had detectable IgG to *Ae. aegypti* salivary gland extract (SGE) prior to mosquito feeding. The team observed induction of neutrophil degranulation and recruitment of skin-resident dendritic cells and M2-macrophages, followed by T cell priming and regulatory pathways along with a shift to a Th2-driven response and CD8+ T cell activation.

Prior animal/in vitro studies have shown that arthropod inoculation of vector saliva, and/or concomitant blood-feeding by an arthropod, can immunomodulate the host antiviral response in the skin and periphery, with dampening of the proinflammatory responses, recruitment of infection-susceptible cells to the bite site, stimulation of autophagy activation, and an induction of neutrophil infiltration.

While the study is largely descriptive, the clinical study and assays are well-executed and the human data are unique and important. The Discussion does an admirable job of synthesizing the array of assays and immune responses to generate a clear narrative that weaves together the sequence and interacting aspects of host responses that develop after mosquito biting. The findings are relevant to better understanding transmission of important vector borne pathogens and can be useful to conceive potential interventions.

While the authors highlight findings that are consistent with earlier preclinical studies, I would have liked to learn where the data here diverge from earlier animal or in vitro studies, if at all. Do the data diverge at all? The fact that the population has pre-existing *Aedes* antibodies and apparently extensive exposure to *Aedes* makes one suspect that some responses here would be modulated from what one might observe in animal or in vitro studies (or in populations with minimal exposure to *Aedes*). In that regard, the evidence that salivary gland extract inhibits specific pro-inflammatory responses of mosquito-bitten skin cells fits with this general idea.

Thanks for the considerations, We have added the paragraph below to the discussion lines 439-455 as follows:

“Our results refine and extend the knowledge obtained from mouse models of mosquito biting. For example, in mouse models, increased frequencies of dendritic cells, total macrophages and neutrophils have been detected after mosquito bites^{21,53,54}. In our human participants, we also observe an increase in the frequency of these cells, indicating that the mouse models of mosquito biting recapitulate the early innate events in human skin. However, we expand these data by showing that the observed increased frequencies are due to increases in M2-like macrophages, which are activated, and increases in Langerhans cells at 4 hours post-bite in our human participants. Additionally, we substantially increased our knowledge on human T cell responses to mosquito saliva by observing CD8 T cells exhibiting a high activation phenotype, while CD4 T effector memory cells shift from a Th1/Th17 phenotype to a Th2/Th17 phenotype 48 hours post-bite. Saliva or salivary proteins from ticks and sand flies have shown ability to inhibit proliferation

of both murine and human cells.^{55,56} We here confirm the inhibitory effects of SGE to skin cells recruited during a mosquito bite. Nevertheless, as previously showed in individuals from an endemic area of leishmaniasis,⁵⁷ the immune responses to saliva of arthropods also triggers a pro-inflammatory response highlighted by the upregulation of pathways related to interferon genes. This mix of anti- and pro-inflammatory responses need to be further evaluated in animal models of mosquito bite exposure as well as clinical field settings.”

Specific issues

1. How were biopsy sites selected? Non-random selection (eg, selecting the largest lesions) could bias the pattern of the responses seen.

We thank the reviewer for this question. This point is valid which is why we only selected ‘discrete’ or ‘distinct’ bites as opposed to bites that ‘coalesced’ into one another. We added this information to the methods as below:

“A total of five 3-mm skin punch biopsies were performed on each individual: one biopsy from bitten skin per time point (30 minutes, 4 hours, and about 48 hours after feeding is completed) for a total of three ‘bitten’ biopsies, and two from NSK collected at baseline. Biopsy sites were selected based on the 3 bite sites with the largest diameters at 30 minutes after feeding. We tattooed the area surrounding those 3 bite sites with a permanent marker and randomly picked one of the three distinct sites for 30 min, 4 hours and 48 hours biopsies. Less than 1cc of 0.2% lidocaine without epinephrine was delivered via 30-G needle each chosen ‘bite’ at the time of biopsy and 3-mm biopsy was performed on ‘bite’ adjacent to needle puncture.”

2. Please standardize the colors when the same pathways are seen in the two panels of Fig 3D—it is confusing when you show the same pathways as different colors in upper and lower panels. Conversely, different pathways should NOT share the same color in the two panels.

We have modified figure 3D to match the color schemes as suggested.

3. Fig 4B—panel is labelled as “Total DC” but y axis indicates these are measured as “% of DC”. Please clarify meaning here of the measurement indicated by y axis—as written, “total DC” should consistently be measured as 100% for “% of DC”, no? One value is >100%-- how does that happen? What is the denominator here?

We thank the reviewer for his remark. We have refined our gating strategy (see Figure S3). Total DCs are now defined as the sum of the percentage of CD45⁺CD3⁻CD123⁺, CD45⁺CD3⁻CD1c⁺ and CD45⁺CD3⁻CD207⁺ and reported as % of CD45⁺. In addition, as per comment of reviewer 2 and 3, we report frequencies of all subpopulations in Figure 4 as % of CD45⁺ cells and have added Figure S5 where we report the frequencies as % of total live cells.

Reviewer #2 (Remarks to the Author):

The main strength of this report is its comprehensive nature and the use of relevant human skin sourced from individuals who live in regions of the world that are affected by mosquito-borne virus disease. The report contains a detailed description of these human skin responses to mosquito biting, from a clinical dermatology, histological, whole transcriptome analysis and cellular response perspective.

Corrections required

“First in human evaluation” is not accurate, as there have been previous longitudinal studies on human skin inflammatory responses to mosquito biting e.g. a recent example;
<https://pubmed.ncbi.nlm.nih.gov/30353912/> This sentence should be modified to clarify and the ref should be added to e.g. line 67.

We apologize if our title seems presumptuous. There are many studies where humans have been exposed to mosquito bites experimentally, dating back to the 1946 when Mellanby (Nature; 1946) first described the immunomodulatory effects of mosquito saliva on the human host. However, the main difference to our study is that all of the prior works have all been observational (such as the recommended reference), NOT interventional. We make this distinction in the title given that we were required to file this study in clinicaltrials.gov as a Phase I Interventional trial, the first of its kind to give us actual molecular and cellular insight in ‘the mosquito bite reaction in humans.’ Further, the reviewer is correct that longitudinal studies such as Oka et al. are certainly informative, and perhaps a limitation of our study is that it is cross-sectional (NOT longitudinal) given our skin observations are limited to 48 hours. An alternative title is “Evaluation of the human cutaneous innate and adaptive immunomodulation by mosquito bites” but we wanted the readers to instantly realize this was an interventional trial by employing ‘first-in-human.’

Line 55 -56. There is little evidence to suggest that mosquito biting leads to immune suppression within the time frame relevant for responses to arbovirus infection (lymphotoxin responses take many days to weeks) . The overwhelming majority of studies in mice show that the host response to mosquito biting is a pro-inflammatory event that does not suppress innate immune responses to virus. This should be rephrased.

We thank the reviewer for this remark. We have adapted this sentence in the introduction to reflect more accurately what has been reported previously in the literature.

“These perturbations result in altered cytokine production profiles¹¹⁻¹³, promotion of recruitment of infection-susceptible cells to the bite site^{14,15}, stimulation of autophagy activation⁵, and an induction of neutrophil infiltration to the local inflammation site¹⁵ amongst other consequences.”

Ref 12 and 32 appear to be duplicates.

We have corrected this oversight.

The paper refers to “extended data” which I believe are supplementary files? These should be re-worded accordingly. Also, they are out of order (e.g. Extended data 3 is referenced in the text before extended data 2).

The word “extended” has been replaced by Supplementary. The order of the supplementary data has been corrected so it follows the correct order of appearance in the text.

Figure 2

Excellent description and summary of human population, with post exposure to dengue and Aedes salivary proteins measured. However, it is not clear in figure 1 legend how bites were biopsied. As described, it gives the impression that an individual bite site was biopsied three times (30 mins, 4 hours and 48 hours). If this is the case, then the act of biopsy would alter inflammation at the site for later time points. If the authors mean that separate bite sites were biopsied for different time points, then this should be made clear.

We thank the reviewer for this comment. We have modified the figure legend to make clear biopsies were taken from distinct bite sites for the different time points. We also added information on the methods that is pertinent.

Figure 1 Legend now reads:

“Thirty Cambodian participants were enrolled in the study. Each patient underwent a mosquito feeding (5 females Aedes aegypti) for ten minutes. Each volunteer underwent a 3mm biopsy from a mosquito bite site at 30 minutes, 4 hours, and 48 hours. Distinct bite sites were biopsied at each time point. Additionally, two 3mm biopsies of normal skin (NSK) were taken from the opposite forearm. Blood was collected to obtain serum.”

Given Reviewer 1 had a similar comment, we wanted to make it clear that we biopsied separate bites (not coalesced bites) that were discrete such that the biopsy itself would not alter results from future biopsies. Methods line 445-447 now reads:

“Biopsy sites were selected based on the 3 bite sites with the largest diameters at 30 minutes after feeding. We tattooed the area surrounding those 3 bite sites with a permanent marker and randomly picked one of the three distinct sites for 30 min, 4 hours and 48 hours biopsies.”

Figure 3

The gene expression analyses is well done, correctly powered and highlights interesting features in host response. This analysis suggests that innate immune gene induction is present by 4 hours (eg cxcl8, and other neutrophil associated functions) and that genes associated with adaptive immune response and wound healing are present by 48 hours. This is perhaps not completely unexpected but importantly does suggest findings from mouse models, that have similar early induction of neutrophil responses, are broadly applicable to human infection. That mouse models of mosquito biting recapitulate these seen here in human skin is very helpful for the research community to know and should be more explicitly stated.

We thank the reviewer for this remark and have added this more clearly to the discussion in response to similar remark from Reviewer 1 (see above). We agree this is the most critical point of the work.

Line 182 – this is wrong figure – do you mean Supplementary figure 6?

Thank you for this correction, we have modified it accordingly.

Figure 4.

- It is not clear why there is no description of neutrophils numbers. Especially as the transcriptome analysis reveal neutrophils associated pathways as the most upregulated. This is a deficiency that cannot be easily corrected as fresh human skin samples would be required. Especially for earlier time point, if histological sections exist, these should be stained for a neutrophil specific marker and numbers of neutrophils quantified in an unbiased fashion.

We agree with the reviewer that the inclusion of neutrophil markers would have been very informative, in hindsight. However, biopsies were processed for gene expression analysis and flow cytometry at the same time, and we were limited in the number of fluorophores. Nevertheless, we can define a population of mostly neutrophils by selecting $CD45^+ CD11b^+ CD14^- CD56^- CD117^-$, where we include myeloid cells that are not lymphocytes, monocytes/macrophages, NK cells or mast cells. However, these population likely includes few DCs and other myeloid cell types that we cannot exclude based on our panel design. We have added this information as figure S6 and in the text line 199-201 as below:

Unfortunately, we did not include neutrophil markers to our panel, alternatively, we selected $CD45^+ CD11b^+ CD14^- CD56^- CD117^-$ cells, that comprises mostly neutrophils. We can detect a significant accumulation of these neutrophils at 4 hours post-bite compared to NSK.

Regarding the histological sections, we had those committed to future studies using digital spatial profiling techniques.

Figure S6. Changes in myeloid cells CD45+CD11b+CD14-CD56-CD117- frequencies. (a) Gating strategy defining the population of myeloid cells excluding lymphocytes, monocytes/macrophages, NK cells and mast cells. (b) Change in the frequency of neutrophils – defined as CD45+CD11b+CD14-CD56-CD117-. Statistical analysis were performed with Wilcoxon signed-rank test. Bars indicate median.

- It is not clear what 'frequency' refers to in each figure panel. Is this frequency of cell to total live cells, or of 'parent', or of CD45+ve cells? As these are whole tissues analyses, frequency is most accurately made in reference to per live cell (e.g. Langerhan cells per live cell), as increases in total cd45+ cell count could affect reported frequencies. Please clarify and re-analyse as required. This may explain why there is a reported apparent increase in Langerhans cells. This is unexpected as these cells tend to become rapidly depleted in skin following inflammatory stimulus and then are only replaced in the medium term (e.g. days to weeks in mouse models). This should be reflected in the description of the results.

We currently modified figure 4 to show the reported frequencies of the cell populations based on total number of leukocytes (CD45+) (panel b, c, d, f, h). In addition, we have added a new supplementary figure (Figure S5), reporting the cell populations as frequency of live cells. By reporting as percentages of live cells we see no differences in the frequencies of any innate cell subset, even though the trends remain. We have added this to the results section.

We added this information to the text lines 213-216 as follows:

In order to evaluate if changes in total leukocytes (CD45⁺) were affecting the reported frequencies, we analysed the changes of the innate cell population within total live cells. Here, similar results were observed (Supplementary Figure S5).

- There is also an apparent increase in M2 macrophages. Is this likely? (no). The authors should report and discuss this carefully. As time required for M2 macrophage differentiation is far longer than 4 hours.

We thank the reviewer for this comment. We modified the text as follows beginning at line 341:

“The observed increase in CD163+CD14+ cells at 4 hours is what we believe to be an initiation of the differentiation of monocytes and macrophages into M2-skewed macrophages given that full macrophage polarization does not occur as early as 4 hours²⁷⁻²⁹.”

Figure 5

- As for figure 4, it is not clear what frequency is being referred to. As whole tissue analysis, this should be a frequency compared to number of live cells (not frequency of parent).

We thank the reviewer for this remark. what we report in Figure 5 are changes within each T cell compartment (CD4 or CD8). Since in this figure we are not comparing T cells to other cell populations, we believe that our approach is acceptable. Nevertheless, we have added in Figure S7 the same analysis based on the number of live cells. Similar results were observed.

We have added the sentence to the text lines 250 as below:

“Results were similar when analyzing the changes of T cell populations within all live cells (Supplementary Figure S7).”

Figure S7. Frequency changes in CD4+ and CD8+ cell populations as percentages of total live cells. Statistical analysis was performed with Wilcoxon signed-rank test. Bars indicate median and interquartile range.

- There is no evidence that these CD8 T cells are exhausted. This reference should be removed and cells simply referred to as “activated”.

We thank the reviewer for this remark and have removed exhausted from the text.

- Most viruses have successfully disseminated via lymph, and infected tissues remote from the bite site, before this 48 hour time point (albeit as informed by mouse studies e.g. see <https://pubmed.ncbi.nlm.nih.gov/21147918/>). Therefore, any response in the skin at this later time point may have questionable consequence for defining outcome to infection. This should be stated clearly and conclusions made changed accordingly.

We thank the reviewer for this remark and have added this point to our discussion, Lines 385-89.

“In case of arbovirus infection, virus will replicate locally in the skin, and the local skin immune responses will direct the course of the systemic host immune response. At 48 hours post-bite, arboviruses will also have disseminated to draining lymph nodes⁴², hence, the contribution of the localized Th2 polarization at this time point on disease outcome remains to be determined.”

Figure 6

- This is an interesting figure, but it lacks an important control and some key experimental details. It would be most informative to compare data from mosquito-bitten skin to normal resting skin, ideally from mosquito bite unexperienced individuals, or at least from sites that are unlikely to have

experienced biting. That way it would be possible to conclude whether mosquito biting influences capability of tissue to respond to further activation with saliva or SGE.

- The figure also lacks experimental detail
 - o time of stimulation with PMA or SGE?
 - o The amount of SGE used?
 - o How SGE was derived?

We thank the reviewer for this remark. Normal resting skin from unexperienced individuals were not possible, given that our study was conducted in an Aedes endemic region and that people have the propensity to be bitten anywhere particularly given children and adults are not fully clothed all the time. Comparison to NSK from the endemic individuals was shown in figure S9. We have added an extra line in the results section describing these data. We have added the experimental details to the figure legend and to the methods section.

- It is not clear to what degree SGE replicates exposure of cells to saliva from mosquito. For example, SGE will contain many cellular components that are not present in mosquito saliva. It is perhaps not surprising that a cellular homogenate has a detrimental effect on cultured cells. Quantification of indicative cell survival analysis should be done to show that cells are not dying, which would explain decreased cytokine expression.

To prepare our SGE, the salivary glands have been extensively sonicated and ultra-centrifugated and after that only the supernatant collected as the SGE. The resulting solution has been tested by SDS-PAGE and Coomassie staining for quality control and that displays the expected salivary proteins. Moreover, we can confirm that there is no effect on cell death since during our optimization experiments we performed flow cytometry staining with live/death staining on the stimulated cells and did not observe any meaningful change in cell viability. Additionally, as shown in figure S9, there are no differences in cytokine production between the unstimulated condition (NSK) and the SGE treatment only condition. We updated this section in the Methods

- Due to these issues with SGE, this experiment would be best done using saliva obtained by forced salivation from mosquitoes. If not possible, this limitation should be discussed.

Lines 411 now reads:

“One consideration is that we used SGE, rather than mosquito saliva as forced salivation did not provide sufficient material, but our preparations minimize the cellular components of the salivary glands.”

Discussion

- The first paragraph contains much needed methodological detail. This should be removed and inserted into the results section to clarify approach.

We thank the reviewer for this remark although our intent was to summarize the study’s findings as the first paragraph of the discussion, which we feel is a standard approach to academic writing. Methodological details were added to the results section that we hope clarify our approach for the Reviewer. We are open to further changes, but do not quite understand exactly what edits would help the Reviewer.

- Much of the language is too generalised and not specific enough. E.g. “innate immune cells” entry at 30 minutes is not accurate (line 285-286). The analysis of the most relevant innate immune cell type (neutrophils) was not done (e.g. mouse studies show rapid entry of neutrophils in response to a variety of arthropod biting e.g. <https://pubmed.ncbi.nlm.nih.gov/18703742/> <https://pubmed.ncbi.nlm.nih.gov/27332734/> . The cell types studies should be specified.

We have modified the discussion accordingly to be more specific when discussing cell populations.

Yes, we have studied Pinggen et al paper in great detail when designing many of our timepoints and gating strategies. Reviewer 1 had a similar comment, and we refer you to the answer above that expounds on RNAseq data showing involvement of neutrophil-dominant pathways and new insights from flow cytometry.

Reviewer #3 (Remarks to the Author):

Focus on the skin as earliest frontline of host/pathogen interaction is interesting as it orchestrates downstream immune reactivity. The in principle descriptive data provide an comprehensive overview of genes, cell types and pathways in response to Aedes mosquito saliva in previously exposed volunteers over time. The paper is well written.

Major comments

The context of the study is placed in the background of Aedes transmitted viral diseases. The study design is limited to the responsiveness to uninfected mosquitoes in dengue-endemic participants. Comparison with infectious bites would have substantially increased relevance but obvious ethical constraints will prohibit such a study. An alternative design would have been studies in non-exposed Aedes volunteers (e.g. from non-endemic areas) complemented with infected bites using the existing controlled human dengue infection models. This should be mentioned and discussed.

*We agree with the reviewer on the value of studying ‘infected’ bites and we hope that ethical and regulatory hurdles may be cleared in the future in order to safely study the phenomena of ‘infected bites since all current CHIM use needle inoculation. **Discussion has been added at Line 369 onwards.***

Intensity and frequency of previous mosquito bites will have a great influence and likely affect responses. Included adult participants do have a history of Aedes exposure and Aedes aegypti salivary IgG enzyme-linked immunosorbent assays is used. Conclusion is made that there are no intra-group variations. However, there is no reference to the value and/or validation of this assay in relation to intensity or duration of mosquito exposure. Age, housing and geo-location are established parameters affecting exposure and biting rates.

We use arbitrary ELISA units, which has now been added in detail on correcting factors and reporting to Methods. In terms of validation of the ELISA assays for Aedes aegypti SGE, we can refer the reviewer to the JID paper (Manning et al 2021) where this assay was used and clearly fluctuated up and down with rainy and dry seasons as mosquito exposure increased and

decreased respectively. Further, intensity of antibody levels using this assay correlated to disease risk which also gives us confidence that the assay corresponds to mosquito exposure. Age, housing, and geo-location certainly affect biting rates, but traditional entomological indices tend to be wildly unpredictable in terms of equating to disease risk (as an indirect measure of mosquito exposure). Given no significant differences between the groups were detected, we wanted to point out that the conclusions drawn from each modality (RNAseq vs flow for example) would not be greatly influenced by a huge discrepancy in prior exposure to mosquito saliva given that all individuals had grown up in Cambodia in the same area and at that point in time, had a similar level of anti-SGE antibody.

The authors indicate that the findings can be “leveraged to develop novel therapeutics and vector-targeted vaccine”. Please elaborate and make suggestions.

Several studies in mouse models (Refs.11-16 in the paper) have showed that saliva increases severity of dengue infection. Since mosquito saliva is always included with the pathogen, if we can block responses to these proteins, we could tilt back the scale in the host’s favor. This was the proof of concept in our first-in-human Anopheles saliva peptide Phase I vaccine trial that utilized a bioinformatic approach to identify candidate peptides (Manning et al, Lancet 2020). Our recent publication at JID (Manning et al 2021) hints on that, where we find that anti-Aedes saliva IgG is higher in dengue inapparent cases compared to apparent dengue infections. The case for vaccination is mentioned several times, but in terms of saliva-targeted therapeutics we can only speculate at the moment. Imiquimod has shown anti-viral activity in the skin (Bryden et al, 2020) via macrophage activation. In terms of therapeutics and prophylactics specific to the saliva-mediated immunological cascade, we can speculate that blocking the chemokine CXCL8 (increased 75x at 4hrs post bite compared to NSK) could preclude the recruitment of neutrophils helping the host control the initial virus dissemination. Separately, we can consider expanded application of newly identified targets to diseases outside of arboviruses – for example, better understanding triggers of M1/M2 polarization can be leveraged to develop therapeutics for patients with psoriasis and other autoimmune disorders. We have added a few lines to the discussion as follows starting at Line 442:

These newly identified genes, cell types, and processes of human cutaneous immunity can be leveraged in the development of vector-targeted vaccine candidates or novel therapeutics such as those targeting the reversibility of functional macrophage polarization or chemokine blockades that may have application in other inflammatory disorders¹. Such new pathogen-agnostic strategies have shown promising results in recent years from mosquito saliva-based vaccinations to topical applications like imiquimod post-bite that activated skin macrophages in order to limit pan-viral activity⁵⁵⁻⁵⁸. Building on these findings, the next generation of countermeasures against vector-borne disease will require nuanced and comprehensive understanding of skin immunity to both pathogen and vector saliva alike.

125: “These data indicate that time post-bite rather than inter-individual differences drive unique patterns in gene expression”. I disagree and would limit the claim to the conclusion that inter-individual differences do contribute to bite-driven patterns.

We are intrigued by this comment from the reviewer. This sentence is driven by the PCA results that are based on the gene expression patterns. The main finding was that instead of finding clusters of subjects, we observed clustering by time points. The Hierarchical clustering may help the review visualize this by showing that at the top 7 out of 10 donors clustered by the 48h time points, but only 2 out 10 (D18 and D13) at the bottom of the graph clustered by inter-individual gene patterns.

Fig 4+5: Legend: Please clarify “% of immune cells” compared to what? Provide the absolute numbers of the 100% values in the legend.

We thank the reviewer for raising this issue, which was also pointed out by reviewer 2. We currently adapted figure 4 to show the reported frequencies of the cell populations based on total number of leukocytes (CD45+) (panel b, c, d, g, j). We report in Figure 5 as frequencies of CD4 or CD8 as we are interested in changes within each T cell population, respectively. We have added a new supplementary figure (Figure S5, S7), reporting the cell populations as frequency of live cells in order to reflect better total cell composition.

Fig 4: The 4hrs data indeed show significant changes compared to 30' or NSK time points at group level but range is often large. When analyzing individual data in the different figures, do volunteers cluster in relative strong and weak responders when comparing this different cell-types? Relation with dengue or mosquito previous exposure?

We have performed correlation analysis between responders (defined based on bite size reaction) and the frequencies of different cell populations at the different time points. Significant correlations are reported in the supplementary material (figure S4).

Fig 6: Cell number and composition of the stimulation assays is not provided. This complicates and limits interpretation of the data. Moreover, only 7/10 volunteers are presented. In vitro responses to Dengue antigens could have been informative but were not included.

We have added these details to the materials and methods section and to the figure legend of figure 6. We included 9 individuals for this analysis, one individual out of the group of 10 individuals showed high background cytokine production in the negative control and was therefore omitted. We agree that in vitro responses to DENV antigens would be interesting to assess, however, cell numbers to perform the in vitro assays were limited.

333: Reference is made to injected microbiota in other diseases while saliva is considered a major trigger (319). Since the mosquitoes used were not sterile, could injected bacteria play a role in the generated data and responses of individual volunteers? Please comment

This is an excellent point and one that we would very much like to study. Recent animal studies do suggest that microbiota play a role; however, we did not investigate this question here. In the future, we would like to do a study where we can either use irradiated (e.g. sterile) mosquitos compared to our locally reared mosquitos as a case-control on the same volunteer or abrogate the microbiota by administering antibiotics to either the mosquito OR even to the volunteer. Anopheles mosquitos who fed on antibiotic-laced blood from children INCREASED the susceptibility of mosquitos to Plasmodium infection because of a disturbance of gut microbiota (Gendrin et al, Nature Communications, 2015) – this is a very important question that also needs to be studied for Aedes and arboviruses.

Because laboratory strains differ in gut microbiota than wild mosquitos, we did strive to use recent field-derived strains that would not have gone through the same bottlenecks to consider the gut microbiota as close to 'real world' as possible.

Minor comments

Is there a particular rationale for the selected time points e.g. hypersensitivity?

Yes, we chose these timepoints in order to capture potential innate responses (30 min and 4 hours) and then adaptive responses but geared toward hypersensitivity at 48 hours as you suggest, specifically because of previous studies that show the role played by delayed-type hypersensitivity with sand fly saliva and leishmaniasis. (Oliveira et al, J Inv Dermatology 2013).

REVIEWERS' COMMENTS

Reviewer #1 (Remarks to the Author):

Manning and colleagues have satisfactorily addressed my comments on their manuscript. Readers will be interested in this Phase 1 interventional, within-individual controlled (normal skin versus bitten skin) study of mosquito bites in a target population of individuals at high risk of vector borne disease, with its extensive cataloguing of innate and adapted immune responses. This study establishes a precedent upon which to build in this field, examining mosquito bites (and in future infective mosquito bites) of human subjects under controlled study conditions to better understand the molecular and immunological consequences of this pivotal event in transmission of numerous pathogens. The authors have done a masterful job of weaving a narrative and visuals to describe the complex interplay of events orchestrated by the immune system in response to the world's deadliest creature, the mosquito.

A few minor points and an amateur editor's red pen--

Lines 110-111: "Additionally, two 3mm biopsies of normal skin (NSK) were taken from the opposite forearm." Please indicate here that these were collected at baseline before mosquito biting (if that is the case).

Fig 2 a, two left panels: Images are obscured by a series of parallel horizontal white lines.

Line 146 "31 downregulated genes" versus Line 149 "one gene at 4 hours and 27 genes at 48 hours" do not add up to the same total number. Check your numbers.

Line 118: "associated with associated with"

Line 213: "inIn order"

Line 262: "Frequencies of CD8 T cells naive (CCR7+CD45RA+), central memory (CCR7+CD45RA-), effector memory (CCR7CD45RA-) and TEMRA (CCR7-CD54RA+) frequencies." The "frequencies" are too frequent.

Line 337 "Indeed" and Line 339 "Indeed". Ditto the above comment.

Line 278: "This data suggests" should be "These data suggest"

Line 353: "Two recently identified Ae. aegypti salivary proteins (NeSt1 and AgBr1)..." is an incomplete sentence lacking a main verb.

Line 418: "the reducing effect of SGE on cytokine production" is better stated as "the effect of SGE to reduce cytokine production"

Line 445: "other inflammatory disorders58com" needs to lose the "com".

Reviewer #2 (Remarks to the Author):

The authors have revised the manuscript and answered all comments satisfactorily. Considering how hard it is to source these samples and undertake this research in samples from relevant human cohorts, this is a great effort and would benefit the research community through its publication.

A few minor comments/typos are listed below.

The lines numbers stated in the rebuttal are not the same as those in the manuscript. This made finding the edits/changes a little difficult. It would also have been beneficial to see what text had been changed e.g. through use of highlighting the new text with a font colour.

Starting on line 421;

“Our results refine and extend the knowledge obtained from mouse models of mosquito biting. For example, in mouse models, increased frequencies of dendritic cells, total macrophages and neutrophils have been detected after mosquito bites
21,53,54”

Ref 53 appears to be a mistake (referring to a similarly named review article). It should be ref 12.

Clive McKimmie, University of Leeds, UK.

Reviewer #3 (Remarks to the Author):

The comments have been satisfactorily addressed and adaptations have improved the manuscript

Response to reviewer comments

NCOMMS-22-10786A

Reviewer comments	Author response
Reviewer #1	
Lines 110-111: "Additionally, two 3mm biopsies of normal skin (NSK) were taken from the opposite forearm." Please indicate here that these were collected at baseline before mosquito biting (if that is the case).	(From Figure 1 caption) Suggested edits made
Fig 2 a, two left panels: Images are obscured by a series of parallel horizontal white lines.	Fixed and improved picture per editor
Line 146 "31 downregulated genes" versus Line 149 "one gene at 4 hours and 27 genes at 48 hours" do not add up to the same total number. Check your numbers.	Now line 122 This is now "4 genes at 4 hours"
Line 118: "associated with associated with"	Now line 155 Corrected
Line 213: "inIn order"	Now line 171 Corrected
Line 262: "Frequencies of CD8 T cells naive (CCR7+CD45RA+), central memory (CCR7+CD45RA-), effector memory (CCR7CD45RA-) and TEMRA (CCR7-CD54RA+) frequencies." The "frequencies" are too frequent.	(From Figure 5 caption) Suggested edit made
Line 337 "Indeed" and Line 339 "Indeed". Ditto the above comment.	Now line 257 Suggested edit made
Line 278: "This data suggests" should be "These data suggest"	Now line 208 Corrected
Line 353: "Two recently identified Ae. Aegypti salivary proteins (NeSt1 and AgBr1)..." is an incomplete sentence lacking a main verb.	Now line 273 Corrected
Line 418: "the reducing effect of SGE on cytokine production" is better stated as "the effect of SGE to reduce cytokine production"	Now line 339 Suggested edit made
Line 445: "other inflammatory disorders58com" needs to lose the "com".	Now line 366 Corrected
Reviewer #2	
Starting on line 421;	Corrected

“Our results refine and extend the knowledge obtained from mouse models of mosquito biting. For example, in mouse models, increased frequencies of dendritic cells, total macrophages and neutrophils have been detected after mosquito bites 21,53,54” Ref 53 appears to be a mistake (referring to a similarly named review article). It should be ref 12.	
--	--